# Towards Specialized Web Agents Using Production-Scale Workflow Data

## Abstract

Large Language Model (LLM) agents are rapidly improving to handle increasingly complex web-based tasks. Most of these agents rely on general-purpose, proprietary models like GPT-4 and focus on designing better prompts to improve their planning abilities. However, general-purpose LLMs are not specifically trained to understand specialized web contexts such as HTML, and they often struggle with long-horizon planning. We explore an alternative approach that fine-tunes open-source LLMs using production-scale workflow data collected from over 250 domains corresponding to 6 billion tokens. This simple yet effective approach shows substantial gains over prompting-based agents on existing benchmarks—our WorkflowAgent achieves state-of-the-art performance on Mind2Web and substantially improves the baseline task success rate from 37.2% to 51.3% on WebArena. We further perform detailed ablation studies on various fine-tuning design choices and provide insights into LLM selection, training recipes, context window optimization, and effect of dataset sizes.

## 1 Introduction

Large language model (LLM) agents have advanced significantly in web navigation. They can carry out user-specified tasks in multiple steps by reasoning on their own what actions to take and what external resources to interface with. Recent studies (Zheng et al., 2024; Lai et al., 2024; Zhang et al., 2024) have shown that, with better planning and exploration strategies, LLM agents can independently solve various web tasks ranging from simple navigation, such as locating a specific Wikipedia page, to more complex operations, such as booking flights or restaurants.

Despite these improvements, the performance of existing web agents on research benchmarks remains significantly below human levels (Deng et al., 2023; Zhou et al., 2024; Drouin et al., 2024). One possible reason is their dependence on *general-purpose* LLMs. Indeed, all top-performing agents like WebPilot (Zhang et al., 2024), AWM (Wang et al., 2024b), and SteP (Sodhi et al., 2024) rely on prompting proprietary models like GPT-4 (OpenAI, 2024a). These general-purpose LLMs are not optimized for interpreting web contexts such as HTML or accessibility trees; their pretraining and alignment processes do not address navigation-related challenges; and their proprietary nature presents a major obstacle in adapting them to web environments via continual training.

In this work, we explore an alternative approach by fine-tuning open-source LLMs with a large set of real-world web workflow data[1] to develop *specialized* web agents (Figure 1). Through extensive experiments, we show that this approach not only boosts the web understanding and planning abilities of LLMs, achieving state-of-the-art results on various benchmarks, but also allows us to develop agent models significantly smaller than proprietary LLMs, drastically reducing the serving costs.

To achieve these results, we first collect a set of proprietary workflow data representing action sequences executed by real users in real web environments. This dataset encompasses a large and diverse spectrum of websites (over 250 domains and 10,000 subdomains), task objectives, task difficulty, and task length. Each step in the workflow features not only the raw HTML-DOM of the website but also a comprehensive documentation of the action, including action description in

---

[1]Due to privacy concerns, we restrict access to our proprietary dataset. However, we will release a version of WorkflowAgent trained on open-source datasets (Deng et al., 2023). We will also release our data preprocessing and model fine-tuning code.

Figure 1: **Left:** Most existing LLM web agents are built on top of general-purpose, proprietary models like GPT-4 and rely heavily on prompt engineering. Their performance is enhanced by leveraging external planning, reasoning, and memory modules. **Right:** We explore an alternative way to develop specialized agents by fine-tuning open-source LLMs using a large set of high-quality, real-world workflow data. This significantly boosts agent's navigation and planning capacity, enabling it to outperform proprietary models with a smaller LLM backbone, thereby reducing serving costs.

natural language, mouse or keyboard operation, and the CSS selector of the target HTML element. We reformat the data into a next-step prediction formulation and fine-tune a set of open-source LLMs via the parameter-efficient LoRA (Hu et al., 2022). After preprocessing and reformatting, our training dataset contains more than 6 billion tokens.

With access to this production-scale dataset, we develop WorkflowAgent, the first family of specialized, single-stage LLM agents capable of directly generating the next step based on the website's DOM and action history. This is in contrast with previous fine-tuned agents that require multiple stages to produce an action, e.g., first narrowing down to a set of target element candidates and then selecting one from the candidates (Deng et al., 2023). WorkflowAgent significantly outperforms existing GPT-4-based and multi-stage agents. Notably, our 7B-parameter model achieves state-of-the-art performance on Mind2Web (Deng et al., 2023) with an over 50% step success rate and a nearly 10% task success rate. These numbers substantially surpass the typical 30% step success rate and 1-2% task success rate seen in existing prompting-based agents. On the end-to-end task execution benchmark WebArena (Zhou et al., 2024), WorkflowAgent boosts the task success rate from 37.2% to 51.3%, marking the highest performance among all published, text-only LLM agents.

Beyond the empirical results, our work also provides several insights valuable for future web agent research: (1) we show that direct fine-tuning on highly structured inputs (HTML-DOM) is feasible and can improve the agent's ability in identifying the correct target; (2) we identify an effective HTML preprocessing strategy that balances between preserving essential information and minimizing context length; (3) we provide a thorough analysis on various design choices in fine-tuning, such as LLM backbone and context window selection; (4) we illustrate how fine-tuning improves agent performance as dataset size increases.

Our work highlights the potential of building web agents via specialized fine-tuning with production-scale data. This approach not only improves agents' capabilities relative to prompt-engineered alternatives, but also reduces inference costs due to the smaller sizes of open-source LLMs. While our work focuses on studying the effect of fine-tuning, WorkflowAgent can be extended to leverage more sophisticated search or memory modules (Koh et al., 2024; Wang et al., 2024b), combined with existing planning frameworks (Yao et al., 2022; Madaan et al., 2023; Shinn et al., 2023), or integrated into multi-modal web agent systems as the text model (Wang et al., 2024a). We view WorkflowAgent as an important step towards developing AI assistants and fully automated agents for real-world web applications.

## 2 RELATED WORK

**Prompting-based agent frameworks.** The majority of web agent works reuse existing LLMs and propose different prompting strategies to improve action prediction. One line of research focuses on exploiting previous experience via self-feedback (Sun et al., 2023) or in-context demonstrations (Fu et al., 2024; Zheng et al., 2024; Wang et al., 2024b; Ou et al., 2024). A separate line of work centers around encouraging exploration by including external evaluators (Pan et al., 2024), using synthesized instructions (Murty et al., 2024), or applying more sophisticated search algorithms like stack (Sodhi et al., 2024), best-first tree search (Koh et al., 2024), or Monte Carlo Tree Search (Zhang et al., 2024). Chain-of-Thought (Wei et al., 2023) or ReAct (Yao et al., 2023) prompting have also

been used (He et al., 2024). Despite the research efforts, these prompting methods rely heavily on the quality of the LLM used. Open-source models such as LLaMA (Dubey et al., 2024), Code LLaMA (Rozière et al., 2024), and Flan-T5 (Chung et al., 2022) generally underperform proprietary models like GPT-4. However, fine-tuning proprietary LLMs can often be costly and challenging, as it is restricted to being done through APIs. This implies an opportunity for enhancing open-source LLMs to match or outperform proprietary agents.

**Fine-tuning-based web agents.** Compared to developing better reasoning and planning frameworks, comparatively less attention has been given to optimizing the LLMs themselves to better handle web environments. Due to the difficulty of directly generating a single target element from the raw HTML, which often contains thousands of elements, existing work mostly focuses on multi-stage prediction. MindAct (Deng et al., 2023) proposes a two-stage pipeline that first uses a small LM to filter the web elements and then uses a more powerful LM to select from the filtered elements in a multi-choice question answering format. Both LMs can be fine-tuned using the Mind2Web dataset. WebAgent (Gur et al., 2023) uses HTML-5 to first process the HTML and then fine-tunes a 540B Flan-UPalm to generate code for controlling web pages. More recently, AutoWebGLM (Lai et al., 2024) trains a single ChatGLM3 6B (GLM et al., 2024) using a combination of curriculum learning, reinforcement learning, and rejection sampling fine-tuning. Despite the complicated training and inference procedures, these methods often underperform agents that prompt GPT-4. In contrast, our work shows that given sufficient high-quality workflow data, fine-tuning a single LLM can achieve strong performance. We note that the newly released OpenAI o1 (OpenAI, 2024c) can be viewed as a specialized agent with a complicated planning framework. Nonetheless, we show in Section 4.1 that WorkflowAgent outperforms o1-preview by a large margin on our proprietary dataset. Moreover, while none of the training details for o1 have been released, our work provides valuable insights into data preprocessing and fine-tuning.

Beyond the aforementioned work, there is an earlier line of research that fine-tunes LLMs for HTML inputs (Gur et al., 2022; Nakano et al., 2022; Liu et al., 2023). However, their primary application is question-answering tasks, such as answering "could sunflowers really track the sun across the sky", and they cannot be used to generate a sequence of actions based solely on the user objective.

Lastly, we note that an emerging line of research has committed to developing multi-modal web agents that use screenshots along with HTML observations. Examples include CogAgent (Hong et al., 2023), SeeClick (Cheng et al., 2024), WebVoyager (He et al., 2024), and AWA 1.5 (JaceAI, 2024). However, our current version of WorkflowAgent focuses exclusively on text-based inputs due to the lack of extensive, high-quality paired data and effective visual preprocessing schemes. Thus, we do not include comparisons with the aforementioned multi-modal methods in our experiments and leave developing multi-modal WorkflowAgent as future work.

## 3 METHOD

In this section, we first overview the general setup of solving web-based tasks with LLM agents. Then, we detail our proposed method to develop specialized agents from open-source LLMs.

### 3.1 GENERAL SETUP

We consider solving web-based task as a sequential decision-making process guided by a high-level objective. For each task, the user first specifies an objective and a starting web page. Then, at every step, the agent outputs an action based on the task objective, the current web page, and the history. Formally, denote the user objective as $q$. The web environment is governed by a transition function $T$ that can evolve over time. The agent is instantiated by a language model $L$. At each time step $t$, the agent observes $o_t$ produced by the environment state $s_t$ and observes the history $h_t = H(o_{1:t-1}, a_{1:t-1})$. It outputs an action $a_t = L(q, o_t, h_t)$, which is executed in the environment, and the state changes correspondingly $s_{t+1} = T(s_t, a_t)$. This iterative process stops when the agent issues a stop signal, or a task termination condition is met, such as we have reached a predefined maximum number of steps.

For single-modal, text-only agents, the observation $o_t$ typically consists of the website's URL, the HTML-DOM (Object Model for HTML, which defines HTML elements and their properties, methods, and events), and potentially the accessibility tree (a representation that can be understood by

assistive technologies like screen readers). Since the raw HTML-DOM is often long and contains redundant structural information, most methods employ preprocessing and pruning strategies, which could be as simple as retaining a fixed set of HTML tags and attributes or more complex ones like LLM-based element ranking and filtering (Deng et al., 2023).

The action $a_t$ emulates the keyboard and mouse operations available on web pages. The most general action space in existing work consists of element operations, such as clicking, typing, and key combination pressing; tab actions, such as opening, closing, and switching between tabs; navigation actions, such as going forward and backward in the browsing history (Zhou et al., 2024).

As discussed earlier, previous web agent work focuses on presenting useful demonstrations through $h_t$ or iteratively revising $a_t$ to improve the quality of the predicted next step. In contrast, we explore whether we can improve the model $L$ itself by learning from a vast amount of data and incorporating more information into $o_t$, such as the natural language description and HTML representation of a action. We detail our approach in the next section.

## 3.2 WORKFLOWAGENT: SPECIALIZING WEB AGENTS THROUGH FINE-TUNING

### 3.2.1 COLLECTING PRODUCTION-SCALE DATA

We collected a large set of real-world proprietary data through a workflow documentation software that streamlines the creation of step-by-step guides to achieve web-based tasks. The software allows users to record their interactions with the web through a browser extension and converts the interactions into well-annotated instructions, which can be then customized to specific business needs. Our dataset consists of everyday workflows in common web application domains, encompassing customer relationship management (CRM) tools like HubSpot and Salesforce; productivity tools like Notion and Calendley; social platforms like Facebook and LinkedIn; shopping sites like Amazon and Shopify; and many others.

Each workflow features a high-level user objective and a step-by-step documentation of the action sequence to achieve the task. The objective spans a wide range of topics, such as "add a user in a Salesforce" or "invite someone to manage Facebook ad accounts". Each step contains the following information: the current web page's URL, raw HTML-DOM, a natural language description of the action performed, the type of action, and the autogenerated CSS selector to identify the action target. There are three types of actions in the dataset:

- *mouse_click_action*: click at an element
- *keyboard_sequence_action*: type a sequence of characters to an element
- *keyboard_combination_action*: press a set of keys together (e.g., hotkey like ctrl+c)

Note that there is no scroll actions in our action space since all elements are already fully accessible in the captured data. This is because we capture the full DOM from a system perspective, which inherently includes the entire webpage as observed from the backend. This method differs from user-centric data collection, where only the elements within the visible browser viewport are captured.

To ensure the quality of the data, we remove workflows with invalid selectors, i.e., the selector cannot be used to locate a target element in the DOM. We also remove non-English workflows to reduce dataset complexity and enable us to explore English-only LLMs like Mistral 7B (MistralAI, 2023). The resulting dataset is at production scale: using raw data collected over a two-month period, we are able to extract workflow data from more than 250 domains and 10,000 subdomains with an average task length of 11 steps, which correspond to about 6 billion training tokens. This large-scale, high-quality, real-world dataset is unmatched in prior web agent research.

Since this dataset is collected from real users and might contain sensitive and confidential information, it will not be released to the public to protect user privacy. The dataset is solely for research purposes and has been anonymized to prevent the identification of any individual.

### 3.2.2 PREPROCESSING

For WorkflowAgent, we consider an observation space consisting mainly of the URL and HTML-DOM. Specifically, HTML-DOM provides agents with all structural and content information about

the web page that are essential for generating the next step and long-term planning. For instance, while a drop-down menu may not be visible on the website before expansion, the agent can detect the menu items from the DOM and determine whether to click and expand it. We do not use accessibility tree to develop WorkflowAgent because it may lose information about the HTML elements, such as the drop-down items, and does not generalize across different browsers and devices.

Given our observation space, a subsequent problem is that the DOM can be quite long and exceed the context window of prevailing open-source LLMs. To reduce the DOM sizes, we propose a pruning algorithm that maintains the essential structure and content while eliminating redundant or disruptive elements that could hinder the LLM's understanding. Specifically, we first use the BeautifulSoup library (Richardson, 2007) to remove non-essential components such as metadata, CSS, and JavaScript. Then, we utilize a tag-attribute white list to retain useful tag level information like retaining interactive elements. Since some attribute values can contain random character sequences that do not provide useful information, we propose a novel detection method that removes the attributes with character-to-token-ratio smaller than 2, i.e., $\frac{len(s)}{len(tokenizer(s))} < 2$, where $s$ denotes the value string. Intuitively, if each character in a string is encoded using a separate token, it is highly likely that the string is not semantically meaningful. Lastly, we remove the comments and extra whitespaces to clean up the DOM. After pruning, we assign each tag in the HTML with a unique ID by traversing the HTML tree from bottom to top. More details about preprocessing and analysis on the tokenizer-pruning method can be found in Appendix A.1.

We restrict the action space of WorkflowAgent to the three types of operations specified in Section 3.2.1. To preprocess the action sequences, we rewrite each step into five lines as follows:

```
1.
Description: Click the "Menu" button to browse all food options
Action: mouse_click_action
Node: 832
Target: <svg class="open-hamburger-icon" node="832" role="img">
```

The first line represents the current time step. The second line is the natural language description of the action, which can help LLMs to learn about the rationale behind applying a specific action. The third line is one of the three operations in the action space. The fourth line is the unique ID assigned to the target element. The last line details the HTML tag and attributes, which can be directly obtained from the processed DOM.

For the history, we consider only previous actions, omitting previous observations due to the extensive length of the DOMs. That is, $h_t = a_{1:t-1}$. Therefore, at each step, WorkflowAgent will be given the task objective, URL, HTML-DOM, and all previous actions in the aforementioned five-line format. Its goal is to output the next action $a_t = L(q, o_t, a_{1:t-1})$ that helps complete the task. In Appendix A.3, we provide an example of a full workflow.

Lastly, during our inspection, we find that 10% of the action descriptions in the dataset are not informative (e.g., "click here"). In these cases, we use GPT-4o (OpenAI, 2024b) to regenerate the action description from screenshots. We provide the prompt as well as examples of the regenerated action descriptions in Appendix A.4.1.

### 3.2.3 FINE-TUNING WITH LoRA

After preprocessing, we divide the dataset into two splits. The test set comprises of 1200 workflows with diverse objectives and domains. We use the remaining workflows as the training data to adapt LLMs via standard supervised fine-tuning. Note that for each example, the label is a single next-step instead of all remaining steps needed to complete the task. The agent is trained to generate all information in the five-line format described above, including the natural language description.

To reduce fine-tuning cost, we opt for the parameter efficient method LoRA (Hu et al., 2022) instead of full fine-tuning, since we have not observed significant performance gain by updating more parameters. We also follow previous work (Zhao et al., 2023) to fine-tune the layernorms in addition to the LoRA adapters. Based on empirical observations, we set the fine-tuning epoch to 2, effective batch size to 32, LoRA rank to 64 and $\alpha$ to 128. We use a cosine scheduler with 30 warmup steps and a learning rate of 1e-4.

Table 1: Performance of different LLMs fine-tuned on 1B workflow tokens on the test split of our proprietary dataset. We highlight the best results for small/medium/large models. EM is short for Exact Match. *Qwen2 57B is fine-tuned at a 29K context window and evaluated on a subset of samples due to compute constraints.

| Model | # Params | Before Fine-Tuning | | After Fine-Tuning | |
|---|---|---|---|---|---|
| | | EM (%) | Calibrated EM (%) | EM (%) | Calibrated EM (%) |
| Mistral-7B-Instruct-v0.3 | 7B | 3.89 | 5.13 | 19.92 | 26.31 |
| Qwen2-7B-Instruct | 7B | 6.06 | 7.92 | **29.34** | **38.72** |
| Llama-3.1-Instruct-8B | 8B | 1.42 | 1.88 | 28.34 | 37.42 |
| Qwen2.5-14B-Instruct | 14B | 8.79 | 11.6 | **31.76** | **41.89** |
| Codestral-22B-v0.1 | 22B | 4.53 | 6.08 | 31.11 | 41.25 |
| Mixtral-8x7B-Instruct-v0.1 | 56B-A12B | 7.35 | 9.82 | 28.38 | 37.49 |
| Qwen2-57B-A14-Instruct | 57B-A14B | 5.72 | 7.51 | 31.02 | 40.10 |

### 3.2.4 EXPLORING THE DESIGN SPACE

There are multiple design choices for WorkflowAgent that might affect the prediction accuracy, fine-tuning cost, and inference latency. We focus on three aspects and perform detailed ablation studies to find out the optimal modeling and training configurations.

**Pretrained LLM Selection.** Intuitively, the quality of a fine-tuned web agent should be relevant to the quality of the pretained LLM. We identify two axes that are crucial to performance—model architecture and model size—and explore seven open-source LLMs spanning these axes: Llama 3.1 8B (Dubey et al., 2024), Mistral 7B (MistralAI, 2023), Mixtral 8x7B (MistralAI, 2024b), Qwen2 7B (Yang et al., 2024), Qwen2 57B (Yang et al., 2024), Qwen2.5 14B (Yang et al., 2024), and Codestral 22B (MistralAI, 2024a). We fine-tune these models with 1 billion training tokens and evaluate their performance on the test split of the dataset we collected.

Given that many of the evaluated LLMs have a maximum context window of approximately 32K, and the processed DOM can exceed this limit, we divide the DOM sequentially into chunks that fit into the context window. For fine-tuning, we use the chunk containing the correct target, but for evaluation, we use the last chunk since the target's location is not known beforehand. When evaluating at a 32K context window, 25% of the test data do not have the correct target tag in the DOM, i.e., these tasks are unachievable. Thus, we compute two metrics for evaluation: (1) exact match (EM) measures the model's ability to select exactly the same HTML tag as the ground truth; (2) calibrated exact match (Calibrated EM, or CEM) measures the percentage of correct target predictions where the target tag was present in the truncated HTML

Table 2: Ablations on context window length.

| Model | Context | EM (%) | CEM (%) |
|---|---|---|---|
| Qwen2 7B | 32K | 29.34 | 38.72 |
| Qwen2 7B | 65K | 31.42 | 36.22 |
| Qwen2.5 14B | 32K | 31.76 | 41.89 |
| Qwen2.5 14B | 65K | 33.96 | 39.15 |

Table 3: Ablations on dataset size. All settings are trained and evaluated with Qwen2-7B-Instruct and 32K context window.

| # Train Tokens | EM (%) | CEM (%) |
|---|---|---|
| 1B | 29.34 | 38.72 |
| 3B | 32.65 | 43.06 |
| 6B | 34.96 | 46.42 |

DOM, i.e., it is EM on the set of examples where the observation contains sufficient information to complete the task. As we scale the context window, these two metrics converge. DOM chunking presents a limitation due to relatively small context windows, which can introduce noise into evaluations. Therefore, effectively extending the context window or developing inference strategies that avoid the need to truncate long observations is a crucial next step for this work.

We report the performance of different LLMs before and after fine-tuning in Table 1. Notably, for all models, specialized fine-tuning drastically increases the prediction accuracy. Among the models with <10B parameters, Qwen2 outperforms both Mistral 7B and Llama 3.1. We observe performance gains as model size increases. For example, the calibrated EM for Qwen2 57B is higher than its 7B counterpart. Mixtral 8x7B outperforms Mistral 7B by a large margin as well. However, fine-tuning larger models is significantly more resource-intensive—while Qwen2 7B can be fine-tuned using 8 H100 GPUs in just one day, Qwen2 57B takes over a week using the same hardware configuration. Larger models also incur longer inference times and require multiple GPUs even at a 32K context length. Among the seven LLMs, Qwen2 7B strikes a balance between prediction accuracy, fine-tuning and inference costs. We thus use it as the default backbone for WorkflowAgent.

**Context Window Length.** We evaluate the models with 65K context window to add additional context and increase the rate of solvable tasks (Table 2). On both Qwen2 and Qwen2.5, scaling the context window from 32K to 65K leads to approximately 2% performance boost for Exact Match but approximately 2.5% performance drop for Calibrated Exact Match. We hypothesize that this performance degradation might be due to rotary position embedding (Su et al., 2021) and the fact that it becomes harder to pick the correct target given twice as many options to choose from. Besides, we note that using 65K context window increases the inference time by approximately $4\times$ in practice.

**Dataset Size.** Lastly, we are interested in understanding the effect of fine-tuning dataset size on the agent's performance. To this end, we sample our training set without replacement into smaller subsets and fine-tune Qwen2 7B on them. Results are shown in Table 3. Plotting on a log-linear scale, we observe that there is a roughly 2% performance boost when we double our dataset size.

To sum up, using our proprietary dataset, we study the effect of LLM backbone, context window, and dataset size on the agent performance. We find that (1) scaling parameter count generally improves prediction quality, but the latency and training time of large LLMs can be prohibitive; (2) using longer context window boosts model performance on EM but increases the inference time significantly; (3) training with more tokens is helpful. Based on these insights, we use Qwen2 7B fine-tuned on the full 6B-token dataset at a 32K context window as the final version of WorkflowAgent. The results shown in later sections are based on this model. Since WorkflowAgent only has 7B parameters, it is much cheaper to serve at inference time than large-scale proprietary models.

## 4 RESULTS

We evaluate WorkflowAgent on three web datasets. We first consider the next-step prediction setting, where performance is evaluated only on a single next step. We show that WorkflowAgent not only outperforms various general-purpose LLMs on our proprietary dataset but also achieves state-of-the-art on the public benchmark Mind2Web (Deng et al., 2023). Then, we move to the end-to-end task completion benchmark WebArena (Zhou et al., 2024) and show that WorkflowAgent augmented with GPT-4o achieves top performance among all existing agent systems.

### 4.1 PROPRIETARY DATASET

To study whether specialized fine-tuning is indeed beneficial, we first compare the performance of WorkflowAgent with general-purpose baselines on our proprietary test data. We consider the non-fine-tuned Qwen2 7B, GPT-4o, and GPT-4o mini. We use in-context demonstrations to prompt them to generate actions in the same five-line format as defined in Section 3.2.2. All OpenAI baselines in this work follow the prompt in Appendix A.4.2.

Results on the full 1200 test workflows are shown in Table 4. We note that WorkflowAgent significantly outperforms the proprietary GPT-4o and 4o mini. This shows the benefit of specialized fine-tuning over using general-purpose LLMs. Moreover, while the non-fine-tuned Qwen2 performs extremely poorly, fine-tuning with our dataset boosts its performance by nearly $6\times$, which highlights the importance of domain-specific data.

Table 4: Comparing specialized WorkflowAgent with general-purpose, non-fine-tuned baselines on the full test set.

| Model | EM (%) | CEM (%) |
|---|---|---|
| Qwen2 7B | 6.28 | 8.20 |
| GPT-4o mini | 12.60 | 13.26 |
| GPT-4o | 15.24 | 16.02 |
| WorkflowAgent | **34.96** | **46.42** |

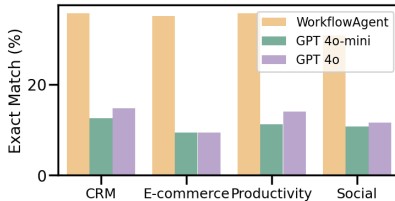

Figure 2: EM comparison between WorkflowAgent and OpenAI models on different domains.

Table 5: Comparing WorkflowAgent with OpenAI baselines on 500 test samples. Since OpenAI baselines are evaluated at a longer 128K context window, they have a smaller gap between EM and CEM.

| Models | Context | EM (%) | CEM (%) |
|---|---|---|---|
| o1-mini | 128K | 17.40 | 18.32 |
| o1-preview | 128K | 22.60 | 23.79 |
| GPT-4o mini | 128K | 13.80 | 14.53 |
| GPT-4o | 128K | 16.60 | 17.96 |
| WorkflowAgent | 32K | **44.6** | **53.86** |

We also plot the Exact Match metric for four types of commonly seen domains, including customer relationship management (CRM) tools, E-commerce platforms, productivity tools, and social

Table 6: WorkflowAgent achieves state-of-the-art on Mind2Web. EA is short for element accuracy, $AF_1$ is short for action $F_1$, and SR is short for success rate. We note that the three categories are based on increasing level of domain generalization difficulty. However, since we do not train on Mind2Web data, our performance is similar across different test sets.

| Method | Uses M2W train set? | Cross-Task | | | | Cross-Website | | | | Cross-Domain | | | |
|---|---|---|---|---|---|---|---|---|---|---|---|---|---|
| | | EA | $AF_1$ | Step SR | Task SR | EA | $AF_1$ | Step SR | Task SR | EA | $AF_1$ | Step SR | Task SR |
| **Multi-Stage, Multi-Choice QA** | | | | | | | | | | | | | |
| MindAct (Flan-T5$_B$) | ✓ | 43.6 | 76.8 | 41.0 | 4.0 | 32.1 | 67.6 | 29.5 | 1.7 | 33.9 | 67.3 | 31.6 | 1.6 |
| MindAct (Flan-T5$_L$) | ✓ | 53.4 | 75.7 | 50.3 | 7.1 | 39.2 | 67.1 | 35.3 | 1.1 | 39.7 | 67.2 | 37.3 | 2.7 |
| MindAct (Flan-T5$_{XL}$) | ✓ | 55.1 | 75.7 | 52.0 | 5.2 | 42.0 | 65.2 | 38.9 | 5.1 | 42.1 | 66.5 | 39.6 | 2.9 |
| AutoWebGLM (ChatGLM3) | ✓ | - | - | **66.4** | - | - | - | 56.4 | - | - | - | 55.8 | - |
| AWM-offline (GPT-4) | ✓ | 50.6 | 57.3 | 45.1 | 4.8 | 41.4 | 46.2 | 33.7 | 2.3 | 36.4 | 41.6 | 32.6 | 0.7 |
| MindAct (GPT-4) | ✗ | 41.6 | 60.6 | 36.2 | 2.0 | 35.8 | 51.1 | 30.1 | 2.0 | 21.6 | 52.8 | 18.6 | 1.0 |
| AWM-online (GPT-4) | ✗ | 50.0 | 56.4 | 43.6 | 4.0 | 42.1 | 45.1 | 33.9 | 1.6 | 40.9 | 46.3 | 35.5 | 1.7 |
| **Direct Generation** | | | | | | | | | | | | | |
| Flan-T5$_B$ Fine-Tuned | ✓ | 20.2 | 52.0 | 17.5 | 0 | 13.9 | 44.7 | 11.0 | 0 | 14.2 | 44.7 | 11.9 | 0.4 |
| HTML-T5-XL | ✓ | **60.6** | 81.7 | 57.8 | 10.3 | 47.6 | 71.9 | 42.9 | 5.6 | 50.2 | 74.9 | 48.3 | 5.1 |
| Synapse (GPT-3.5) | ✓ | 34.0 | - | 30.6 | 2.4 | 29.1 | - | 24.2 | 0.6 | 29.6 | - | 26.4 | 1.5 |
| WorkflowAgent (**Ours**) | ✗ | 54.2 | **90.7** | 52.2 | **10.7** | **58.8** | **88.3** | **57.4** | **10.2** | **58.0** | **86.6** | **56.3** | **8.8** |

platforms (Figure 2). While our agent's performance varies by domain, with a 6% gap between the best performing domain and the worst performing one, we observe that WorkflowAgent consistently outperforms the general-purpose baselines across all of them.

As we were wrapping up this work, OpenAI released o1 (OpenAI, 2024c), a series of specialized models for solving complex tasks in science, coding, and math. Since it has better planning ability, we also include it in our baselines. However, we did not run the o1 models on the full test set due to cost and API call limitations. Instead, we subsample 500 workflows and compare with WorkflowAgent. As shown in Table 5, o1-preview performs the best among all general-purpose baselines. However, WorkflowAgent still outperforms it by a wide margin, highlighting the importance of fine-tuning on real-world web navigation data. It is worth noting that WorkflowAgent only contains 7B parameters and does not require any inference time scaling, whereas most proprietary baselines are typically larger in size and slower at inference time. This makes WorkflowAgent a better choice in terms of accuracy, latency, and cost.

## 4.2 MIND2WEB

Mind2Web (Deng et al., 2023) is a text-based dataset for assessing the navigation ability of web agents across different tasks, websites, and domains. Each task features a human demonstration of a real-world workflow, such as booking a hotel on Airbnb. At each step, the agent is asked to predict a single action, consisting of an operation and the target element. Performance is measured by element accuracy, which checks if the correct target is selected; action F1 score, which measures operation correctness like text input; step success rate, which evaluates whether both the target element and the operation are correct; and task success rate, indicating all steps are correct.

The original Mind2Web benchmark reports two sets of baselines: (1) a single-stage, generation-based agent (i.e., fine-tuned Flan-T5$_B$) directly generates the operation and the target based on the full DOM; (2) multi-stage, multi-choice question-answering agents (i.e., the MindAct family) first use a pretrained element-ranking model to filter out 50 candidate elements from the full DOM and then use a separate LLM to recursively select an action from five candidates in a multi-choice question-answering (QA) fashion until one action is chosen. Both sets of baselines are trained using the training data and then evaluated on the test set. Note that direct generation is more challenging than multi-choice QA, and all multi-stage baselines outperform generation baselines by a large margin. Beyond the Mind2Web original baselines, we also consider memory-augmented agents such as AWM (Wang et al., 2024b) and Synapse (Zheng et al., 2024), fine-tuned AutoWebAGLM (Lai et al., 2024) and HTML-T5 (Gur et al., 2023).

WorkflowAgent belongs to the single-stage, generation category. We directly evaluate its performance on the Mind2Web test data *without* using Mind2Web training data for further adaptation. For performance robustness, we call WorkflowAgent five times and use majority vote to select the final generated action. More details about DOM processing and output comparison are in Appendix A.5.

Table 7: Task success rates (SR) on WebArena and score breakdown on five web domains. WorkflowAgent consistently outperforms all considered baselines, often improving the previous-best results by more than 10%.

| Method | LLM | Total SR | Shopping | CMS | Reddit | GitLab | Maps |
|---|---|---|---|---|---|---|---|
| AutoWebGLM | ChatGLM3 6B | 18.2 | - | - | - | - | - |
| AutoEval | GPT-4 | 20.2 | 25.5 | 18.1 | 25.4 | 28.6 | 31.9 |
| BrowserGym | GPT-4 | 23.5 | - | - | - | - | - |
| BrowserGym$_{axtree}$ | GPT-4 | 15.0 | 17.2 | 14.8 | 20.2 | 19.0 | 25.5 |
| SteP | GPT-4 | 33.0 | 37.0 | 24.0 | 59.0 | 32.0 | 30.0 |
| AWM | GPT-4 | 35.5 | 30.8 | 29.1 | 50.9 | 31.8 | 43.3 |
| Tree Search | GPT-4o | 19.2 | - | - | - | - | - |
| WebPilot | GPT-4o | 37.2 | 36.9 | 24.7 | 65.1 | 39.4 | 33.9 |
| Multi-Agent System (**Ours**) | WorkflowAgent + GPT4o | **51.3** | **48.1** | **35.5** | **70.2** | **58.8** | **51.9** |

We report all evaluation metrics in Table 6. WorkflowAgent achieves state-of-the-art performance on Mind2Web. More specifically, for both step and task success rates, we outperform not only the generation baselines but also all multi-stage QA baselines. Our action F1's are significantly higher, which means that WorkflowAgent is good at specifying the content of typing actions. Even though we have not tuned WorkflowAgent on Mind2Web training data, the fact that we outperform Mind2Web fine-tuned models on all except two metrics suggests that WorkflowAgent can generalize across various domains and websites. We attribute this to the diversity and high quality of the workflows in our dataset. Relatedly, the three test sets (Cross-Task, Cross-Website, Cross-Domain) are designed to capture different degrees of domain generalization difficulty. Since we do not train on Mind2Web data, the performance of WorkflowAgent is similar across all three test sets.

While these results are promising, we note that a limitation of static, text-based benchmark is that the ground truth evaluation does not account for different action sequences that could reach the same goal. For instance, to book a flight, one can first enter the destination or first choose the departure date, but the ground truth trajectory only accounts for one possibility. Considering this, we also evaluated WorkflowAgent on a dynamic benchmark WebArena (Zhou et al., 2024).

### 4.3 END-TO-END TASK EXECUTION ON WEBARENA

WebArena (Zhou et al., 2024) features 812 web navigation tasks across five domains: E-commerce (OneStopShop), social forums (Reddit), software development (GitLab), content management (CMS), and online map (OpenStreetMap). Unlike the static Mind2Web, it implements a dynamic environment for agents to interact with and allows for assessing the functional accuracy of action sequences. Since the WebArena environment is implemented to accept only target element IDs specified in the accessibility tree, whereas WorkflowAgent operates on DOM and outputs targets in HTML, we employ GPT-4o to map between the different representations.

More generally, we tackle end-to-end task solving by developing a multi-agent system that utilizes GPT-4o to simulate user interactions with WorkflowAgent: (1) objective refinement: user adds details about the task objective to help complete the task; (2) action generation: based on the current website and action history, WorkflowAgent outputs an action suggestion; (3) action execution: user executes the suggested action, e.g., clicking a button; (4) completeness evaluation: user observes the current state and decides whether the task is completed.

We apply the above pipeline to solve the WebArena tasks. In stage 3, GPT-4o maps the agent's output in HTML to the accessibility tree format, which is then processed by the WebArena environment. To further improve performance, we allow WorkflowAgent to generate multiple actions in stage 2 and select the one with the highest confidence using majority vote and GPT-4o analysis. More details about evaluating WorkflowAgent on WebArena can be found in Appendix A.6.

We compare our performance with all top-performing, text-only agents on the WebArena leaderboard. We note that we do not include Autonomous Web Agent (AWA) 1.5 (JaceAI, 2024) as a baseline because it uses a proprietary system to parse the HTML-DOM and web screenshots, rather than building from the WebArena GitHub. This allows them to have richer observations and bypass the accessibility tree action mapping step. In contrast, WorkflowAgent is single-modal, text-only, and we stick to the original WebArena implementation. That said, AWA 1.5 employs more advanced reasoning, planning, and progress tracking techniques and is the only agent system with a higher average task success rate than ours.

Table 8: We replace WorkflowAgent with GPT-4o in our four-stage pipeline to study how much WorkflowAgent contributes to the performance. The success rates drop significantly for all domains.

| Method | LLM | Total SR | Shopping | CMS | Reddit | GitLab | Maps |
|---|---|---|---|---|---|---|---|
| Single-Agent | GPT-4o | 34.2 | 31.9 | 21.3 | 44.7 | 38.2 | 42.6 |
| Multi-Agent | WorkflowAgent + GPT4o | **51.3** | **48.1** | **35.5** | **70.2** | **58.8** | **51.9** |

Table 9: Task success rates on a subset of WebArena. The numbers after the domains indicate the number of tasks considered. All model are used along with GPT-4o to formulate the multi-agent system. We see that the general trends agree with what we found on our proprietary dataset.

| Agent Backbone | # Train Tokens | Total SR (158) | Shopping (36) | CMS (39) | Reddit (24) | GitLab (33) | Maps (26) |
|---|---|---|---|---|---|---|---|
| Mistral 7B | 1B | 41.8 | 41.7 | 30.8 | 50.0 | 42.4 | 42.3 |
| Qwen2 7B | 1B | 44.3 | 52.8 | 33.3 | 50.0 | 48.5 | 42.3 |
| Qwen2 7B | 3B | 47.5 | 55.6 | 33.3 | 58.3 | 48.5 | **46.2** |
| Qwen2 7B | 6B | **55.0** | **58.3** | **41.0** | **70.8** | **63.6** | **46.2** |

The results are shown in Table 7. Compared with existing text-only baselines, WorkflowAgent augmented with GPT-4o obtains the highest task success rate in all five categories, leading to 14.1% performance improvements in total success rate over the previous-best WebPilot results. In particular, on Reddit and GitLab tasks where the domains are more realistic and thus closer to the ones in our training data, our method demonstrates stronger generalization ability and higher task success rates than in other domains.

To better understand the contribution of WorkflowAgent to the multi-agent system, we perform an ablation study that leverages GPT-4o for all four-stages of the proposed pipeline. As shown in Table 8, using WorkflowAgent consistently outperforms only using GPT-4o, and the GPT-4o-only setting is less effective than existing agents like WebPilot. This shows that our strong performance on WebArena can be mostly attributed to the action generation process of WorkflowAgent. Apart from getting better results, the multi-agent system is cheaper than using GPT-4o alone, as calling WorkflowAgent to generate a next action incurs negligible cost as it is served locally. We follow Agent-E (Abuelsaad et al., 2024) to report the number of API calls for proprietary models. Due to the four-stage pipeline design, our multi-agent system requires 3 GPT-4o calls for each action step (action analysis, action mapping, and completeness evaluation), plus an addition API call at the beginning of each task for objective refinement. This makes our four-stage method more expensive than agent systems that utilize a single API call per step.

We also use WebArena to verify the signals observed in our proprietary test data. To do so, we randomly select a subset of 158 WebArena tasks with non-overlapping objective templates and run ablation studies following the ones presented in Section 3.2.4 to study the effect of LLM backbones and the number of training tokens. As shown in Table 9, on all domains, Qwen2 7B outperforms Mistral 7B, and the task success rate increases as the number of training tokens increases. These trends suggest that improvements on our proprietary dataset lead to even greater improvements on WebArena, further highlighting the advantages of fine-tuning web agents with large-scale datasets.

## 5 CONCLUSION

In this work, we explore how fine-tuning open-source LLMs with high-quality real-world workflow data can benefit developing specialized web agents. We present WorkflowAgent, which consistently outperforms existing methods that prompt proprietary models in various evaluation settings and benchmarks. We also provide empirical insights into data processing and model fine-tuning.

**Limitations and Future Work.** The long-context nature of DOMs presents great challenges in adapting LLMs. In the short term, we aim to enable WorkflowAgent to compare and reason over multiple DOM chunks so that its observation is always complete. This might require integrating a memory component, which could also aid in maintaining context or state across interactions to improve multi-step reasoning. Besides, we currently do not incorporate planning into WorkflowAgent, so its output will be directly used as the next action. However, adding better action selection strategies such as Monte Carlo Tree Search (MCTS) could potentially facilitate online planning and exploration, further improving the agent's decision-making processes in complex scenarios. In the long run, we aim to expand WorkflowAgent's capabilities to handle multi-modal inputs and multilingual content. This would significantly broaden its applicability across different linguistic and visual contexts, making it more versatile and robust in real-world web environments.

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

## A  APPENDIX

### A.1  PREPROCESSING

#### A.1.1  OVERALL PRUNING PIPELINE

We overview the pruning algorithm used for processing the HTML DOM in the paper. The code for preprocessing, chunking the DOM for fine-tuning, fine-tuning, and inference can be found in the supplementary material, which we will release on GitHub later.

```python
def assign_element_id(all_tags):
    for i, tag in enumerate(all_tags[::-1]):
        tag["node"] = int(i)

salient_attributes = { "alt", "aria-role", "aria-label", ... }

def clean_tag(tag):
  for attr in list(tag.attrs):
    if attr in tag:
      if type(tag[attr]) == list:
        tag[attr] = " ".join(tag[attr])
      tag[attr] = str(tag[attr])[:32]
    if len(str(tag[attr])) > 32 and token_ratio(str(tag[attr])) < 2:
      del(tag[attr])
      continue

    if "script" in attr.lower():
      del tag[attr]
      continue

    if attr.lower() not in salient_attributes:
      del tag[attr]
      continue
    elif (tag[attr] == "" or tag[attr] == "none"):
      del tag[attr]
      continue

    if attr in tag:
      if tag.name == "iframe":
        if attr != "node":
          del tag[attr]
  tag = soup.prettify()
  return tag[:tag.find(">")+1]

valid_tags = { 'div', 'body', 'span', 'svg', 'input', 'img', 'p', ...}
code_elements_to_decompose = { 'style', 'script' }

def removing_tags(all_tags):
  count = 0
  for tag in all_tags:
    if tag.name in code_elements_to_decompose:
      tag.decompose()
    elif tag.name not in valid_tags:
      tag.unwrap()
  return all_tags

def generate_output(step_count, desc, kind, element_id, target_html):
  return f"{step_count}.\nDescription: {desc}\nAction: {kind}
  \nNode: {element_id}\nTarget: {target_html}\n"
```

### A.1.2 TOKENIZER PRUNING

In this section, we provide more details on the tokenizer-based detection method to remove random character strings. The rationale behind our approach is based on the observation that typical English words consist of more than two characters. Assuming the token count is $t$ and the character count is $s$, this means that when $t = 1$, $s \geq 2$, leading to $\frac{s}{t} \geq 2$. By setting the pruning threshold to 2 and removing tag attributes with $\frac{s}{t} < 2$, we aim to eliminate strings composed solely of single-character tokens, which are likely to be nonsensical.

In our actual implementation, we employ this technique only for tag attributes with $s > 32$, being more lenient for shorter attributes. To show that this tokenizer pruning strategy is effective and to study the performance across different tokenizers and pruning thresholds, we perform the following experiments.

We take three tokenizers from different models: Qwen2-7B-Instruct, Mistral-7B-Instruct-v0.3, and Meta-Llama-3-8B. For each tokenizer, we vary the pruning thresholds across a set of values: $\{1.5, 1.75, 2, 2.25, 2.5\}$. Note that it is meaningless to study overly small thresholds (e.g., it is impossible to have $\frac{s}{t} < 1$) or overly large thresholds (e.g., $\frac{s}{t} < 3$ could result in the loss of meaningful attributes, as many English words contain three letters). We randomly sample 1000 DOMs from our proprietary test dataset, apply our standard pruning pipeline followed by tokenizer pruning, and then perform three analysis:

- False positives: we use the Python `enchant` library to detect if there are meanful English words within the pruned strings. Note that even though these are actual words, many of them are related to DOM structure and can be safely ignored. Still, we count them as false positives since the tokenizer method is designed to remove random character strings.

- Average $s$ and $t$ for the entire DOM before and after tokenizer pruning: this is for understanding the reduction in content length.

- Lastly, we sort tags and attributes by the frequency of being pruned to identify patterns.

Table 10: Tokenizer pruning analysis.

| Tokenizer | Prune Threshold | False Positive (%) ↓ | Before Pruning (K) | | After Pruning (K) | | |
|---|---|---|---|---|---|---|---|
| | | | $s$ | $t$ | $s$ | $t$ | $\Delta t$ |
| Qwen2-7B-Instruct | 1.5 | 0.025 | | | 221.4 | 77.11 | 2.03 |
| | 1.75 | 0.013 | | | 217.3 | 74.67 | 4.47 |
| | 2 | 0.18 | 224.3 | 79.14 | 215.7 | 73.89 | 5.21 |
| | 2.25 | 0.36 | | | 213.9 | 73.13 | 6.01 |
| | 2.5 | 0.38 | | | 210.0 | 71.63 | 7.51 |
| Mistral-7B-Instruct-v0.3 | 1.5 | 0.012 | | | 219.5 | 87.10 | 3.44 |
| | 1.75 | 0.18 | | | 216.1 | 85.07 | 5.47 |
| | 2 | 0.44 | 224.3 | 90.54 | 212.7 | 83.40 | 7.14 |
| | 2.25 | 0.49 | | | 205.3 | 80.20 | 10.34 |
| | 2.5 | 11.28 | | | 190.3 | 74.44 | 16.10 |
| Meta-Llama-3-8B | 1.5 | 0.0097 | | | 223.1 | 70.60 | 0.84 |
| | 1.75 | 0.012 | | | 218.3 | 67.85 | 3.59 |
| | 2 | 0.035 | 224.3 | 71.44 | 216.8 | 67.09 | 3.43 |
| | 2.25 | 0.023 | | | 215.2 | 66.41 | 5.03 |
| | 2.5 | 0.10 | | | 212.7 | 65.46 | 5.98 |

As shown in Table 10, there is a clear trade-off between precision and context reduction: greater reductions in content length tend to result in higher false positive rates. While different tokenizers exhibit varying sensitivities to the pruning thresholds, a threshold of 2 achieves the most balanced trade-off, which aligns with our intuition. We then list the top-5 tag-attribute pairs most frequently pruned under threshold 2 along with their pruning counts:

- Qwen: ('div', 'class'): 3188, ('span', 'class'): 11426, ('a', 'href'): 8802, ('button', 'class'): 6844, ('i', 'class'): 5010

- Mistral: ('div', 'class'): 5288, ('span', 'class'): 15824, ('a', 'href'): 12948, ('button', 'class'): 7998, ('svg', 'class'): 5871

- Llama: ('div', 'class'): 29559, ('span', 'class'): 8823,('button', 'class'): 5889, ('i', 'class'): 4608, ('svg', 'class'): 2577

Attributes such as 'class' often contain random character strings and are frequently pruned. However, we observe differences in how tokenizers handle the href attribute: both Qwen and Mistral tokenizers tend to prune it away, whereas the Llama tokenizer preserves it, indicating its better capability in tokenizing URLs. Although we currently use the Qwen tokenizer in our preprocessing pipeline to align with the backbone model of WorkflowAgent, the Llama tokenizer can be a compelling alternative for future consideration since it is better at recognizing URLs and producing shorter token sequences.

## A.2 DETAILS ABOUT FINE-TUNING AND INFERENCE

We provide the code for fine-tuning WorkflowAgent and using it at inference time in the supplementary material. The code files include detailed configurations such as learning rate, LoRA configuration, and generation configuration. There is also implementation of our evaluation metrics.

## A.3 EXAMPLE PROMPT AND LABEL FOR WORKFLOWAGENT

```
Objective: Grant delegation access to another user in Gmail settings.
URL: https://mail.google.com/mail/u/0/
Observation: {processed dom}
Step-by-step guide:
1.
Description: Click "See all settings"
Action: mouse_click_action
Node: 254
Target: <button class="Tj" node="254">
2.
Description: Click "Accounts"
Action: mouse_click_action
Node: 2625
Target: <a class="f0 LJOhwe"
↪  href="https://mail.google.com/mail/u/0/?tab=#settings/accounts"
↪  node="2625" role="tab">
3.
Description: Click "Add another account"
Action: mouse_click_action
Node: 1215
Target: 
```

## A.4 OPENAI PROMPTS

### A.4.1 DATA PREPARATION

Below shows the prompt to generate step descriptions.

**Regenerated Action Descriptions.** We provide a few examples of generated action descriptions using GPT-4o.

- "Click on the Submit button."
- "Type in the name of the item."
- "Double-click on the highlighted text."

### A.4.2 PROPRIETARY BENCHMARK BASELINES

Below shows the prompt for all OpenAI baselines. The text is the prepend for every input to which we append the task input with the corresponding objective, URL, DOM, and action history.

```
 You are navigating a webpage to achieve an objective. Given the
 objective, a list of the previous actions, the current action, and a
 screenshot of the current action on the webpage. The objective and
 previous steps are only here to ground the current step, the current
 action and its screenshot are the most useful to your task. Give me
 a concise description of the current action being done on the webpage.
 You should look at the part of the webpage with the red circle, this is
 where the user clicked for the current action. Describe this action
 and ensure your response is in the same format, concise, coherent.
 Use any relevant information in the image to ground the action
 description. Your response should NOT use any json or markdown formatting.
 The response should be a single sentence that starts with an action verb.
 For example, 'Click on the 'SUBMIT' button.'
```

```
 You are an autonomous intelligent agent tasked with solving web-based
 tasks. These tasks will be accomplished through the use of
 specific actions you can issue.
 Here's the information you'll have:
 - The user's objective: This is the task you're trying to complete.
 - The current web page's URL: This is the page you're currently navigating.
 - Part of the current web page's HTML: Each element is assigned in
 descending order with an unique ID, denoted by the attribute \"node\".
 The actions you can perform include:
 - mouse_click_action: click
 - keyboard_sequence_action: type a sequence of characters
 - keyboard_combination_action: press a set of keys together
 (e.g., hotkey like ctrl+c)
 You will generate a step-by-step guide to complete the task based on the
 given information. You will only produce a SINGLE next step.
 Do NOT use additional punctuation, or any markdown formatting.
 The output should be in the following format:
 Description: Click \"Users\"
 Action: mouse_click_action
 Node: 93
 Target: <a node=\"93\" class=\"slds-tree__item-label\">
 Now complete the following task by generating the next step.
 {task input}
```

## A.5 MIND2WEB EXPERIMENT DETAILS

**Data and Label Conversion.** To apply WorkflowAgent to Mind2Web data, we first re-process the provided DOM using the procedure detailed in Section 3.2.2. We store a map between our node ID and the backend ID given in the dataset. Then, we transform the history action provided in the dataset to our 5-line format. After WorkflowAgent generates the next step, we check the backend ID of the provided label and map it to the node ID in our processed DOM. We then compare this label with the target node ID generated by WorkflowAgent. We provide the code for the DOM processing and label conversion process in the supplementary material and will release them later.

**DOM Chunking and Action Generation.** When the DOM length exceeds the 32K context window, we chunk the DOM sequentially and run the prediction workflow on each piece. For each piece of DOM, we call WorkflowAgent five times to obtain five valid actions. We then aggregate all possible actions and select the one with the highest number of appearances. We use the following generation configuration: do_sample=True, top_p=0.95, temperature=0.6.

## A.6   WEBARENA EXPERIMENT DETAILS

### A.6.1   FOUR-STAGE PIPELINE

**Stage 1:** GPT-4o refines the intent. We use the following prompt:

> I have a simple task objective related to [DOMAIN], rewrite it into a single paragraph of detailed step-by-step actions to achieve the task. When revising the objective, follow the rules:
> - Assume you are already on the correct starting website and are logged in.
> - Do not include any newlines, tabs, step numbers in the rewritten objective.
> - Follow the example as much as possible.
> - [IN-CONTEXT DEMONSTRATIONS FOR DOMAIN RULES]
> Here is an example:
> Simple Task Objective:[IN-CONTEXT DEMONSTRATION]
> Detailed Task Objective: [IN-CONTEXT DEMONSTRATIONS]
> Now, rewrite the following objective:

**Stage 2:** We process the environment-generated DOM using our preprocessing procedure. When the DOM length exceeds the 32K context window, we chunk the DOM sequentially and run the prediction workflow on each piece. For each piece of DOM, we call WorkflowAgent multiple times to obtain multiple valid actions. We use the following generation configuration: do_sample=True, top_p=0.95, temperature=0.6. We then aggregate all possible actions, pick the top candidates, and prompt GPT-4o to select the best candidate using the following prompt:

> You are an autonomous agent helping users to solve web-based tasks. These tasks will be accomplished through series of actions. The information you'll have includes:
> - The user's objective
> - The current web page's URL
> - The current web page's accessibility tree
> - Previous steps performed by the user, where each step includes a description of the action and the target web element
> - Several proposed next steps, labeled by "No."
> Your goal is to select the best next step that can complete the task and output this candidate's number, follow the following rules:
> - Do not repeat previous steps
> - Reject candidates with incorrect intentions, e.g., searching for an item different from the one specified in the objective
> - Reject candidates with factual errors, e.g., the description and the chosen web target do not match
> - Only output a single number after to represent the selected candidate but not explanation
> Now analyze the following case:

**Stage 3:** GPT-4o maps the output of WorkflowAgent to accessibility tree format using the following prompt:

You are an autonomous agent helping users to solve web-based tasks. These tasks will be accomplished through series of actions. The information you'll have includes:
- The user's objective
- The current web page's URL
- A snippit of the current web page's HTML
- A snippit of the current web page's accessibility tree
- Previous steps performed by the user
Your goal is to translate a proposed next step, which consists of an action and a HTML element, into the following format:
- 'click [accessibility tree id]': This action clicks on an interactive (non-static) element with a specific id. Note this id is the number inside "[]" in the accessibility tree, not the HTML attribute "node". Brackets are required in the response. For example, a valid response is "click [1234]"
- 'type [accessibility tree id] [content]': Use this to type the content into the field with a specific id in the accessibility tree. For example, a valid response is "type [1234] [New York]". The second bracket should include everything that needs to appear in the textbox, but not only the added content. Do not change the letter case
- 'press [key_comb]': Simulates pressing a key combination on the keyboard (e.g., press [PageDown], press [Enter])
- 'go_back': Return this when the current web page does not contain useful information and the user should go back to the previous web page
When mapping the next step into actions in the above formats, follow the following rules:
- Take the user's objective into consideration, so the action must help complete the task
- Do not repeat previous steps
- Only output a single step in the above format but not explanation
Note also: [IN-CONTEXT DEMONSTRATION OF RULES]
Now analyze the following case:

The action is then returned to the environment for execution.

**Stage 4:** GPT-4o evaluates if the task objective is achieved. For operational tasks, if the task is completed, nothing is returned. For information seeking tasks, if the task is completed, GPT-4o retrieves the answer to the question. The prompt looks like the following:

You are an autonomous agent helping users to solve web-based tasks. These tasks will be accomplished through series of actions. The information you'll have includes:
- The user's task, including a high-level objective and a more detailed illustration
- The current web page's URL and accessibility tree
- Previous steps performed by the user, where each step includes a description of the action and the target web element
You should follow the rules: [IN-CONTEXT DEMONSTRATION RULES]
You will decide whether the task specified by the high-level objective is completed (which means the **last** step of the detailed instruction is completed and the current webpage completes the task) and respond "completed" or "incomplete". If the task requires returning a number or a string and the answer can be obtained in the current webpage, reply "completed, [answer]" where "[answer]" is the number or string. If the task requires finding a webpage and the current webpage satisfies the requirement, reply "completed, [answer]" where "[answer]" is the current URL. Now analyze the following case. First provide the reasonings. Then summarize the answer with "Summary:", followed by "completed" or "incomplete", followed by the answer to the question if applicable. Do not include newlines after "Summary:".

### A.6.2  SCROLLING ACTIONS AND COMBOBOX SELECTION

In our data collection process, we capture the full DOM from a system perspective, which inherently includes the entire webpage as observed from the backend. This method differs from user-centric data collection, where only the elements within the visible browser viewport are captured. Consequently, there is no concept of scrolling in our training datasets since all elements are already fully accessible in the captured data.

However, we recognize the importance of scroll actions in solving WebArena from a user perspective. To address this, before issuing any action to the environment, our multi-agent system includes a viewport check that uses the bounding box position to determine if the target element is within the visible webpage area. If not, the system manually inserts necessary scroll actions to bring the element into view. This ensures accurate interaction with web elements in a typical user scenario.

To handle combox selection, our agent discovers a workaround that bypasses the need for scrolling through comboboxes. Specifically, after clicking on the combobox, it types the name of the desired item in the combobox, which brings the item to the top of the dropdown menu. Then, the agent can simply click the item or press Enter. This approach avoids the need for scrolling and is especially effective in densely populated lists. It improves the task success rate on a large number of Map, Reddit, and GitLab tasks.

### A.6.3  GPT-4O-ONLY SETTING

When we use GPT-4o for stage 2, we use the following prompt:

You are an autonomous intelligent agent tasked with solving web-based tasks. These tasks will be accomplished through the use of specific actions you can issue. Here's the information you'll have:
- The user's objective: This is the task you're trying to complete.
- The current web page's URL: This is the page you're currently navigating.
- The current web page's HTML: Each element is assigned with an unique ID, denoted by the attribute "node".
The actions you can perform include:
- mouse_click_action: click
- keyboard_sequence_action: type a sequence of characters
- keyboard_combination_action: press a set of keys together (e.g., hotkey like ctrl+c)
You will generate a step-by-step guide to complete the task based on the given information. At each step, you can perform only one action to one web element. The output should be in the correct format: a single step consisting of a text description, an action, as well as the node and HTML of the target web element to perform the action. Be coherent, concise, and accurate in your response. Do NOT use any special characters (e.g., "*", "#", etc.) in your response. Follow EXACTLY the format of the response below.
Here is one example:
Description: click "Users"
Action: mouse_click_action
Node: 93
Target: <a class="slds-tree__item-label" node="93">
Now complete the following task by generating a single next step:

### A.6.4  RESULTS

We include the action trajectories of WorkflowAgent on successful tasks in the supplementary material. Below, we provide example action trajectories produced by our agent and explanations to shed light on its decision making process.

Domain: Shopping

Task ID: 466

Intent: Add 2 Hawaiian Bamboo Orchid Roots #zc50 - by Discount Hawaiian Gifts to my wish list.

Actions:

1. click [1585] where [1585] is combobox 'Search' autocomplete: both hasPopup: listbox required: False expanded: False → Click on the search bar.

2. type [1585] [Hawaiian Bamboo Orchid Roots #zc50 - by Discount Hawaiian Gifts ] where [1585] is combobox 'Search' focused: True autocomplete: both hasPopup: listbox required: False expanded: True → Type the target item name.

3. press [enter] → Do the search.

4. scroll [down] → Scroll down the page to find out the target item.

5. click [4859] where [4859] is link '2 Hawaiian Bamboo Orchid Roots #zc50 - by Discount Hawaiian Gifts' → Open the item page.

6. click [6733] where [6733] is link 'Add to Wish List'→ Add item to wish list.

Domain: CMS

Task ID: 244

Intent: Show me the email address of the customer who is the most unhappy with Olivia zip jacket.

Actions:

1. click [108] where [108] is link 'MARKETING' → Click on the "MARKETING" button in the side bar.

2. click [419] where [419] is link 'All Reviews → Clicking the side bar shows more option. We can then click on the "All Reviews" button.

3. click [2195] where [2195] is button 'Reset Filter' → Remove all previous set filters.

4. type [4435] [olivia] where [4435] is textbox '' required: False → In the search box, type "olivia".

5. press [enter]→ Do the search

6. stop [emma.lopez@gmail.com] → Inspect all listed reviews and return the email of the negative review.

Domain: Reddit

Task ID: 403

Intent: Change my reddit bio to "Awesome Prompt Artist"

Actions:

1. click [12] where [12] is button 'MarvelsGrantMan136' hasPopup: menu expanded: False → Click my user name on the top right corner.

2. click [249] where [249] is link 'Profile' → After the drop-down menu shows up, click "Profile".

3. click [886] where [886] is link 'Edit biography' → Click the "Edit biography" button.

4. type [2456] [Awesome Prompt Artist] where [2456] is textbox 'Biography' required: False describedby: user_biography_biography_help → Type the target content.

5. click [2474] where [2474] is button 'Save' → Save the new profile.

Domain: GitLab

Task ID: 293

Intent: Show me the command to clone Super_Awesome_Robot with SSH.

Actions:

1. click [1507] where [1507] is textbox 'Search GitLab' required: False → Click on the search bar.

2. type [1516] [Super_Awesome_Robot] where [1516] is searchbox 'Search GitLab' focused: True describedby: search-input-description → Type the repo name in the search bar.

3. click [2082] where [2082] is link 'Convex Eggtart / Super_Awesome_Robot' → Click on the correct repo.

4. click [2699] where [2699] is link 'Clone' → Click on the "Clone" button.

5. stop [git clone ssh://git@metis.lti.cs.cmu:2222/convexegg/super_awesome_robot.git] → Read the command from the pop-up window.

Domain: Maps

Task ID: 7

Intent: Tell me the full address of all international airports that are within a driving distance of 50 km to Carnegie Mellon University.

Actions:

1. click [35] where [35] is textbox 'Search' focused: True required: False → Click on the search box.

2. type [35] [airport Pittsburgh] where [35] is textbox 'Search' focused: True required: False → Type "airport Pittsburgh" in the search box.

3. stop [Pittsburgh International Airport, Airport Boulevard, Findlay Township, Allegheny County, 15231, United States.] → Return "Pittsburgh International Airport, Airport Boulevard, Findlay Township, Allegheny County, 15231, United States." as the answer.

