# OpenReview forum: "Towards Specialized Web Agents Using Production-Scale Workflow Data"
_ICLR.cc/2025/Conference — ICLR 2025 Conference Withdrawn Submission_

### Official Review · Reviewer_EuX9 · 2024-10-23

**Soundness:** 2
**Presentation:** 3
**Contribution:** 1
**Rating:** 5
**Confidence:** 4

**Summary:**

This paper fine-tunes an LLM with a 6B workflow dataset to play as a web agent. The fine-tuned agent improves performance on some web task benchmarks such as Mind2Web and WebArena, which is, to the best of my knowledge, the first of this kind. However, the author doesn't offer evidence to support their technical novelty, like the dataset used for training (I can understand the privacy concerns), models trained, and the episodes of the agent play, unlike a recent similar work which open-sources all materials to replicate their work: https://arxiv.org/abs/2410.13824. In addition, the agent's good overall performance on WebArena requires the massive intervention of LLMs (gpt4o) to translate the environment, plan, and self-evaluate, so I don't see any points in training a cheaper web LLM but requires more general LLMs as a multi-agent system to perform well.

**Strengths:**

The paper is well written. The web task adaptation of the LLM is novel and seems to be effective, according to their reported figures. The entire pipeline is complete with model comparison, ablation study, and dataset scaling law. The results of the Mind2Web seem promising as the model improves overall performance without adapted training for the dataset.

**Weaknesses:**

I would question the validity of the results. I check your supplementary materials and find the Mind2Web results are missing. The WebArena results are significantly not complete and hard to interpret. I can understand the privacy concerns, so you can not release the datasets, but what about the data collection platform you mentioned in the paper? I don't think the slim "preprocessing.py" code could support that.

In addition, for the real grounding of the web agent, the end-to-end WebArena's evaluation could more faithfully represent the web interaction capability of your agent than Mind2Web. The results are good from your table, but why does your approach intensively require the integration of LLMs like gpt4o? If the agent requires gpt4o to translate the markup language, to plan, and to evaluate so that you can have a better overall performance, then why does your point of "less inference cost" make sense? Isn't this system a combination of many previous techniques?

**Questions:**

- What is the workflow documentation platform you mention in 3.2.1?
- How did you get the action description? From human annotators?
- You seem to have excluded the scroll mouse operation. Why do you have the scroll action in your incomplete WebArena episodes?
- How large is the training dataset? 6B but how many episodes?
- You mention the accessibility tree cannot show whether the element is interactive, but from the WebArena environment using that representation,  it seems to include such information.
- Is your model trained to generate the natural language action description? If not, how could the action description be kept consistent when the agent is implemented?
- The chunk of the observation is unreasonable and unrealistic.
- Exact match is not a good metric to measure the performance of a web LLM. It can only reflect whether the model remembers the dataset patterns well.
- The intensive integration of gpt4o into WebArena task completion is insane. I would call this a multi-agent workflow where only a small portion is replaced with a cheap smaller model.
- Ablation study should be reframed to emphasize how your model contributes to the overall success.
- (Optional) Did you change the interaction script of the WebArena environment? Or otherwise, how could it be possible that you gain 51.9% on the map task without effective combobox selection? And a 70.2% on the Reddit task if the keyword matching in the evaluator is super strict?

---

> ### Author Response · Authors · 2024-11-17
> **Strongly but respectfully disagree with the weaknesses raised in the review**
>
> To address your comments effectively, we have organized our response by grouping similar concerns together. **We have updated the manuscript and supplementary material according to your feedback.** Please check them out. Thank you!
>
> &nbsp;
>
> **Weaknesses: _Mind2Web results are missing. The WebArena results are significantly not complete and hard to interpret._**
>
> We appreciate the reviewer's effort in checking our results. However, we strongly disagree with the questions on our results' validity. We have carefully followed each benchmark's evaluation criteria (note that we will address the concerns regarding WebArena implementation later in this rebuttal). To be even more transparent, we update our work to include:
> - **Detailed Mind2Web results:** We include our agent's outputs for all test data in the updated supplementary material, which clearly demonstrates the high quality of the predicted actions and supports our strong results on Mind2Web. Note that the evaluation code for MindWeb was already provided in the supplementary material.
> - **Explanation on WebArena Results:** Our supplementary material already included the code for the agent system and trajectory demonstrations for all successful tasks. To help understand these results, we select one task for each domain and update Appendix 6.4 to include detailed step-step explanation for these tasks to shed light on why our agent achieves considerably higher task success rates than recent literature.
>
> We hope that these updates will address any concerns about the validity of our results. If you have any other questions, please let us know!
>
> &nbsp;
>
> **Weaknesses: _What about the data collection platform you mentioned in the paper? I don't think the slim "preprocessing.py" code could support that._**
>
> **Question 1: _What is the workflow documentation platform you mention in 3.2.1?_**
>
> **Question 2: _How did you get the action description? From human annotators?_**
>
> The workflow documentation platform we mentioned is a public software tool designed to streamline the creation of step-by-step guides. It allows users to record their interactions with the web through a browser extension and converts the interactions into well-annotated, step-by-step instructions, which can then be customized to   specific business needs. The software tool is closely tied to our institution, so we cannot disclose its name in order to maintain anonymity and adhere to double-blind review policy.
>
> The software records user actions in a structured format, capturing raw HTML, mouse and keyboard interactions, and target HTML elements. Given its goal is to create web workflow instructions that facilitate replication, it requires users to annotate each step thoroughly. These annotations are the `natural language description" we mentioned in the paper, so the action descriptions are derived directly from real human users.
>
> We have updated Section 3.2 to include more details about data collection and output format. We have also updated the supplementary material to include raw data collected by the software from sample workflows we conducted ourselves. The `preprocessing.py` script is for handling the raw data. We also have examples of processed data in Appendix A.3.

---

> > ### Author Response · Authors · 2024-11-17
> >
> > **Weaknesses: _The (WebArena) results are good from your table, but why does your approach intensively require the integration of LLMs like gpt4o? If the agent requires gpt4o to translate the markup language, to plan, and to evaluate so that you can have a better overall performance, then why does your point of "less inference cost" make sense? Isn't this system a combination of many previous techniques?_**
> >
> > **Question 9: _The intensive integration of gpt4o into WebArena task completion is insane. I would call this a multi-agent workflow where only a small portion is replaced with a cheap smaller model._**
> >
> > Thanks for acknowledging the strong performance of our agent. We address your questions from 3 perspectives.
> >
> > **Effectiveness:** For WebArena, our approach indeed relies on two models: our specialized agent and GPT-4o. However, it is essential to note that the most important *action generation* process is handled only by our agent. We show in the ablation study (Table 8) that if we replace our agent with GPT-4o, the performance drops significantly across all domains, which underscores the indispensable role of our agent in the interaction pipeline.
> >
> > **Cost:** We agree that  in the context of solving WebArena, combining our agent and GPT-4o using a four-stage pipeline is more costly  compared to *a single-stage method* where only a single agent is called once to generate each action. However, throughout the paper, our claims are: (1) for action generation, our agent reduces the inference cost compared to proprietary models because it is based on a small open-source LLM; (2) on WebArena, under the same four-stage pipeline, using our agent + GPT-4o is more efficient than using GPT-4o only. We have revised the second–to-last paragraph in Section 4.2 to make this clearer.
> >
> > **Characterization:** Upon reflection and in response to your feedback, we agree that describing our setup as a multi-agent system is a more accurate characterization and have revised Section 4 of the paper accordingly. However, our initial description of the system as an agent and synthetic user interaction remains technically correct. WebArena's guidelines do not preclude the use of multi-agent systems, and our approach fully complies with these standards, so it is reasonable.
> >
> > &nbsp;
> >
> > **Question 3: _You seem to have excluded the scroll mouse operation. Why do you have the scroll action in your incomplete WebArena episodes?_**
> >
> > In our data collection process, we capture the full DOM from a system perspective, which inherently includes the entire webpage as observed from the backend. This method differs from user-centric data collection, where only the elements within the visible browser viewport are captured. Consequently, there is no concept of scrolling in our training datasets since all elements are already fully accessible in the captured data.
> >
> > However, we recognize the importance of scroll actions in solving WebArena from a user perspective. To address this, before issuing any action to the environment, our multi-agent system includes a viewport check that uses `node_info["union_bound"]` to determine if the target element is within the visible webpage area. If not, the system manually inserts   necessary scroll actions to bring the element into view. This ensures accurate interaction with web elements in a typical user scenario.
> >
> > We have added this scrolling mechanism to Appendix A 6.2. Note that we DID NOT alter the WebArena environment setup.
> >
> > &nbsp;
> >
> > **Question 4: _How large is the training dataset? 6B but how many episodes?_**
> >
> > Since our training dataset is private and the agent is intended for business use, we are unable to specify the exact number of workflows in the dataset. That's why we only provided the token count. We appreciate the reviewer’s understanding of not sharing detailed numbers in this context.  We fine-tune our agent on this dataset for 2 epochs.
> >
> > &nbsp;
> >
> > **Question 5: _You mention the accessibility tree cannot show whether the element is interactive, but from the WebArena environment using that representation, it seems to include such information._**
> >
> > Thank you for highlighting this point. We acknowledge that our previous description---accessibility trees “do not show whether an element is interactive”---was imprecise. Our main concern is that for dropdown menus, accessibility trees often do not display the menu items, whereas DOM representation does. We have revised Section 3.2.2 to be more precise about this limitation.

---

> ### Author Response · Authors · 2024-11-17
>
> **Question 6: _Is your model trained to generate the natural language action description? If not, how could the action description be kept consistent when the agent is implemented?_**
>
> Yes, our agent is specifically trained to generate natural language action descriptions. As part of our data collection process, users of our software annotate each step in the workflow with natural language descriptions. Using this real-world data to train our agent enables it to process and generate action descriptions as part of its output.
>
> &nbsp;
>
> **Question 7: _The chunk of the observation is unreasonable and unrealistic._**
>
> We employ chunking for two reasons:
>
> **Efficiency:** Chunking the DOM into smaller pieces reduces the content length, allowing the agent to process each chunk sequentially. This approach is a practical compromise that enables the handling of large DOMs without the need for viewing the entire document simultaneously. Chunking has been employed in previous works: HTML-T5 [1] “truncate the end of extracted snippet to fit within the token budget”; Gur et al. [2] “apply a snippet extraction pre-processing for pages which are too large to fit into the typical LLMs context windows”; and many others. This technique is reasonable and improves the effectiveness of managing long-context web data efficiently.
>
> **Realism:** Chunking effectively simulates the user experience of browsing the web through a single screen at a time. Typically, users can only view a portion of a website at any given moment, much like viewing through the "window" of a browser. This method mirrors the natural limitations encountered during web navigation and is therefore a realistic approach to data processing in our model.
>
> &nbsp;
>
> **Question 8: _Exact match is not a good metric to measure the performance of a web LLM. It can only reflect whether the model remembers the dataset patterns well._**
>
> To clarify, "Exact Match" in our context refers to whether the agent selects the correct web element as the action target. This is determined by comparing the _node ID_ predicted by the agent with the ground truth _node ID_. This metric does not assess whether the agent merely reproduces strings from the input data (i.e., “if the model remembers the dataset pattern”), but rather to evaluate its precision in interacting with web elements.
>
> If we misunderstand your question, please do let us know and we are happy to address it further.
>
> &nbsp;
>
> **Question 10: _Ablation study should be reframed to emphasize how your model contributes to the overall success._**
>
> Thank you for the suggestion. We have revised  Section 4.2 and added a separate table for ablation study (Table 8) to reflect that our model + GPT-4o achieves significantly better performance than using GPT-4o alone. The multi-agent system improves the total  task success rate by 17.1%. This shows that our agent is the main contributor to the strong performance on WebArena.
>
> &nbsp;
>
> **Question 11: _(Optional) Did you change the interaction script of the WebArena environment? Or otherwise, how could it be possible that you gain 51.9% on the map task without effective combobox selection? And a 70.2% on the Reddit task if the keyword matching in the evaluator is super strict?_**
>
> Thank you for noticing the low-level details of WebArena implementation. We DO NOT change the backend implementation or evaluation process of WebArena. However, our multi-agent system effectively overcomes the noted challenges through strategic use of keyboard and output check, resulting in significant performance improvements.
>
> **Combobox Selection:** Our agent discovers a workaround that bypasses the need for scrolling through comboboxes. Specifically, after clicking on the combobox, it types the name of the desired item in the combobox, which brings the item to the top of the dropdown menu. Then, the agent can simply click the item or press Enter. This approach avoids the need for scrolling and is especially effective in densely populated lists. It improves the task success rate on a large number of Map, Reddit, and GitLab tasks.
>
> **Keyword Matching:** we use GPT-4o for action mapping and specify the prompt such that the generated action strictly replicates the user inputs, including the typos in the given objective. This is crucial for Reddit tasks that involve making posts and replies or revising user profiles. For reddit tasks that involve finding an appropriate forum, we leverage both the objective refinement stage and  the action mapping stage to find out the correct forums or replies, which improves our task success rate even under strict evaluation.
>
> We have added the discussion to Appendix A 6.2, and we are happy to answer any further concerns about how specific tasks are achieved!

---

> > ### Comment · Reviewer_EuX9 · 2024-11-23
> >
> > **Keyword Matching**
> >
> > It seems you have defined a system specifically for improving the success rate on the WebArena benchmark, as typically we would expect the llm agent to help correct the typo in the intent. I have the same question as above, that is, how well does your agent generalize to other web tasks/benchmarks if you keep everything the same as you use the agent for WebArena?

---

> > > ### Author Response · Authors · 2024-11-23
> > >
> > > **Keyword Matching**
> > >
> > > We believe it is inaccurate to suggest that “we would expect the llm agent to help correct the typo in the intent.” In fact, what we did in our multi-agent system is that we specified “adhere to the user’s input” in the prompt for GPT-4o. It is reasonable to ask an agent to faithfully process the user’s input without trying to alter it itself, because any modification could result in unexpected outcomes. Moreover, the agent's adherence to user input should not negatively impact its performance on other web tasks or benchmarks, unless those benchmarks specifically require typo correction in user inputs. In such cases, it would be appropriate to adjust the GPT-4o prompt to include a "typo correction" function.   We emphasize all of the prompts discussed above are for GPT-4o but not our agent.

---

> ### Author Response · Authors · 2024-11-21
> **Looking for feedback on rebuttal**
>
> Thank you again for your valuable feedback! At this moment, we hope you have taken the time to consider our responses to your review. Note that to address other reviewer’s feedback, we have further updated the paper to include:
> - More analysis on the pruning method for DOM preprocessing: In Appendix A 1.2, we have added detailed analysis about the tokenizor pruning method mentioned in Section 3.2.2, including ablations on tokenizer choice, pruning threshold, etc.
> - Baseline non-fine-tuned results for proprietary dataset in Table 1: the take away is that without fine-tuning, open-source LLMs tend to perform poorly on web-based tasks since they are not trained to process HTML. However, after our specialized fine-tuning, there is a significant improvement in performance, clearly demonstrating the effectiveness of our approach.
>
> If you have any additional questions or concerns, please let us know so we can resolve them before the discussion period concludes. Otherwise, it would be greatly appreciated if you could raise your score to show that the existing concerns have been addressed. Thank you!

---

> ### Comment · Reviewer_EuX9 · 2024-11-22
>
> Thank you for your clarifications. I have some questions here.
>
> **Effectiveness and Cost**
>
> From the [WebArena leaderboard](https://docs.google.com/spreadsheets/d/1M801lEpBbKSNwP-vDBkC_pF7LdyGU1f_ufZb_NWNBZQ/edit?usp=sharing), the best agent also comes from a private institute (you mention the model is for business use), achieving a success rate of 57.1%, better than your reported 51.3%. In terms of effectiveness, it seems your agent's action generation is not the best in the fine-tuned specialized model track.
>
> In terms of cost, you should consider the application scenes. Most specialized web open-source LLMs, if not all, are intended for low-resource research use. You seem to have dismissed this low-resource setting when training the action generation model, as you significantly increase the LLM calling cost of your system. Take a look at [Agent-E](https://arxiv.org/abs/2407.13032), which shares the "plan, generate action, evaluate" stage but eliminates the markup language translation step. What is the (performance increase)/(cost increase) of these two agents?

---

> > ### Author Response · Authors · 2024-11-23
> >
> > Thank you for your further feedback. Please find our response below.
> >
> > **Effectiveness:** We have clarified in Introduction, Related Work, and WebArena Experiment sections that our method achieves the best task success rate among **text-only** agents. AWA 1.5 from Jace AI indeed reports a success rate of 57.1%. We explicitly discussed it and acknowledged its strong performance in Section 4.3, paragraph 4: _"We compare our performance with all top-performing, text-only agents on the WebArena leaderboard. We note that we do not include Autonomous Web Agent (AWA) 1.5 (JaceAI, 2024) as a baseline because it uses a proprietary system to parse the HTML-DOM and web screenshots, rather than building from the WebArena GitHub. This allows them to have richer observations and bypass the accessibility tree action mapping step. In contrast, WorkflowAgent is single-modal, text-only, and we stick to the original WebArena implementation. That said, AWA 1.5 employs more advanced reasoning, planning, and progress tracking techniques and is the only agent system with a higher average task success rate than ours."_ We believe the above characterization is accurate for both AWA 1.5 and our agent.
> >
> > **Cost:** Thank your for bringing up using API calls to measure cost-efficiency. We added the following discussion to Section 4.3: _"We follow Agent-E  (Abuelsaad et al., 2024) to report the number of API calls for proprietary models. Due to the four-stage pipeline design, our multi-agent system requires 3 GPT-4o  calls for each action step (action analysis, action mapping, and completeness evaluation), plus an addition API call at the beginning of each task for objective refinement. This makes our four-stage method more expensive than agent systems that utilize a single API call per step._"
> >
> > Since Agent-E did not evaluate on WebArena, it is hard to directly compare our agent with it regarding performance increase vs. LLM call increase. Nonetheless, we believe that our current assessment of efficiency on WebArena is accurate: (1) using the same 4-stage pipeline, WorkflowAgent + GPT-4o is cheaper than using GPT-4o alone; (2) we report the API calls per step and acknowledge that due to the system design,  our multi-agent method is more expensive than agents that use a single API call. However, this aspect does not diminish the standalone capacity and practical value of WorkflowAgent.

---

> ### Comment · Reviewer_EuX9 · 2024-11-22
>
> **Observation Chunk**
>
> What if the observation chunk you feed into the model happens to exclude the required web elements for the task ("for evaluation, we use the last chunk since the target’s location is not known beforehand.")? When I mentioned "unreasonable and unrealistic," I would like to emphasize that the agent's performance should not rely on chance introduced by the system.
>
> **Exact Match**
>
> Mind2Web may not be a suitable benchmark for testing the agent's generalizability as it's a fixed environment using the "exact match" as the metric. There are web benchmarks defined on the real-world web that evaluate agents by end-to-end play, such as [WebVoyager](https://arxiv.org/abs/2401.13919). How well does your agent perform there?
>
> **Ablation Study**
>
> How did you construct the ablation study control group? It's weird that the four-staged gpt-4o multi-agent in your experiment setting (34.2%) performs worse than several LLM-based agents on the WebArena leaderboard. It's easy to control the agent to perform worse.

---

> ### Author Response · Authors · 2024-11-23
>
> **Observation Chunk**
>
> - **For our internal dataset:** We introduced the Calibrated Exact Match (CEM) metric specifically to address noise from the pruning of correct targets. It measures prediction accuracy solely when the target is within the observation.  Besides, we use the same preprocessing schemea cross all settings for our ablations, ensuring that the EM and CEM metrics we report accurately reflect the relative capabilities of different models. That said, we acknowledge that DOM chunking, due to limited context windows, can introduce evaluation noise, so we added to Section 3.2.4: _"DOM chunking presents a limitation due to relatively small context windows, which can introduce noise into evaluations. Therefore, effectively extending the context window or developing inference strategies that avoid the need to truncate long observations is a crucial next step for this work."_ Nonetheless, it’s worth noting that most previous work employs truncation to prune DOMs, as mentioned in our initial response.
> - **For Mind2Web:** When the correct target is removed during our DOM pruning process, we count it as an incorrect prediction by our agent.  This is reasonable, because preprocessing is inherently a part of our agent’s workflow. Thus, any errors in removing the correct target are fairly reflected in our evaluations.
> - **For WebArena:** Since there’s no definition of ground truth, any task failure due to the correct target being pruned is captured in the final metrics we report. Again, since DOM preprocessing is part of our agent's functionality, it is reasonable that our evaluations account for preprocessing limitations.
>
> &nbsp;
>
> **Exact Match**
>
> We agree that using a fixed benchmark with exact match as a metric does not fully capture an agent's capabilities. That's why we've chosen WebArena, which is dynamic and interactive. Thank you for suggesting WebVoyager as an additional benchmark We will add this baseline in upcoming work!
>
> &nbsp;
>
> **Ablation Study**
>
> The only difference between the two setups is that in our multi-agent system, we use WorkflowAgent to generate next actions; in the GPT-4o-only system, we use GPT-4o to generate next actions. All the other components are the same. It is unfounded for the review to suggest that we manipulate our agent to perform poorly. The mediocre performance of the GPT-4o-only setting is expected and reasonable, given that it employs a straightforward 4-stage workflow without any external planning modules, such as stack-based policy employed by [1] or self-refine employed by [2]. We also do not use memory modules like AWM [3]. Note that this is because our focus was to evaluate our agent's capabilities, so we did not extensively refine the multi-agent system or integrate more advanced techniques. Consequently, it's likely that the GPT-4o-only agent underperforms existing baselines that utilize prompting-based planning or memory enhancements. However, the fact that replacing the GPT-4o-generated actions with those from our agent significantly enhances performance underscores the effectiveness of our agent. Despite the 4-stage pipeline's limitations, our agent markedly improves overall performance and outperforms existing text-only agents, demonstrating its value.
>
> [1] SteP: Stacked LLM Policies for Web Actions. Sodhi et al.
>
> [2] Autonomous Evaluation and Refinement of Digital Agents. Pan et al.
>
> [3] Agent Workflow Memory. Wang et al.

---

> ### Comment · Reviewer_EuX9 · 2024-12-02
>
> I increased the score as you addressed the concerns I proposed in my first comment, but it seems you have raised more problems in your rebuttal. Specifically, balancing effectiveness and **cost**, chance introduced by **observation chunk**, etc. I appreciate your honesty, but too much to be stuffed into limitations implies less qualified work.

---

### Official Review · Reviewer_Wrtp · 2024-10-31

**Soundness:** 2
**Presentation:** 3
**Contribution:** 2
**Rating:** 5
**Confidence:** 3

**Summary:**

This paper proposes WorkflowAgent, an LLM agent for autonomous web navigation tasks. The authors curate large-scale workflow data in real web environments and use LoRA to fine-tune open-source LLMs like Qwen2-7B-Instruct. The performance of WorkflowAgent is evaluated across three benchmarks, demonstrating generalization abilities and significant improvements, setting new SoTA. Moreover, the authors conduct detailed ablation studies and analyses to verify the effectiveness of the WorkflowAgent. In general, this paper fine-tunes an LLM with a newly curated large workflow dataset on web tasks, revealing the potential of fine-tuning smaller LLMs with large-scale data to outperform much larger LLMs.

**Strengths:**

1. The elaboration of dataset collection and preprocessing in Section 3.2.1-3.2.2 is well-written, clear, and detailed. This can inspire other data collection and processing work in future research.
2. In Section 3.2.4, the authors conduct preliminary experiments to evaluate the impact of model type, window length, and dataset size, well justifying the final settings in the main experiments.
3. Sufficient performance improvement on three different benchmarks show the strong capability and generalizability of the proposed WorkflowAgent.

**Weaknesses:**

1. Novelty. Assisting and automating web navigation tasks with LLMs is an interesting and important topic, and WorkflowAgent indeed shows impressive performance in Section 4’s experiments among text-only methods. However, this paper deploys conventional and common fine-tuning methods (LoRA) on a widely-used open-source LLM (Qwen2-7B-Instruct) for existing scenarios (LLMs for web tasks), making the paper less innovative.
2. Contribution. The large-scale workflow dataset in a real-world web environment plays a vital role in improving WorkflowAgent’s performance, which would have been a primary and important contribution, but it is proprietary and close-sourced due to privacy reasons. In addition, as noted in the footnote of the first page, only WorkflowAgent trained on public datasets will be released, and WorkflowAgent trained on datasets newly curated by authors does not seem to be released. I understand the privacy concerns, but neither the dataset nor the model is open-sourced, it significantly limits the contribution of this paper to the community and makes the experimental results unreproducible. If possible, I kindly ask the authors to consider open-sourcing desensitized datasets or open-sourcing models trained on the newly curated datasets that are used in the experiments for better reproducibility.
3. Motivation. Considering more and more recent studies in this field have utilized MLLMs rather than text-only LLMs to better capture visual information in web tasks, the motivation for choosing a text-only approach needs to be further clarified.
4. Baselines. Some more recent and stronger MLLM web agents also have text-only setting (e.g. WebVoyager_text-only[1]), and they can be compared with the proposed WorkflowAgent as stronger baselines.

[1] WebVoyager: Building an End-to-End Web Agent with Large Multimodal Models

**Questions:**

1. Section 3.2.2 tag attribute filtering: Why are attributes with a character-to-token ratio smaller than 2 considered not semantically meaningful and thus removed? Please elaborate on more details of the intuition here.
2. Would it be possible to open-source the desensitized dataset or only the models trained on the newly curated dataset used in the experiments for better reproducibility?
3. MLLMs can better capture visual information, which can be beneficial in web navigation. What is the motivation for choosing a text-only LLMs-based approach?

---

> ### Author Response · Authors · 2024-11-19
>
> Thank you for your constructive feedback. Before addressing each of your comments individually, we would like to outline key revisions made in the rebuttal:
> - **Regarding data source:** We have added the following clarification to the paper: “our dataset is collected using a public software tool designed to streamline the creation of step-by-step guides. It allows users to record their interactions with the web through a browser extension and converts the interactions into well-annotated, step-by-step instructions, which can then be customized to   specific business needs.” Since the software tool is closely tied to our institution, we cannot disclose its name in order to maintain anonymity and adhere to double-blind review policy. However, to provide a clearer understanding of our dataset, we have included raw data from an example workflow performed by the authors in the supplementary material.
> - **Regarding the DOM pruning method:** In Appendix A 1.2, we have added detailed analysis about the tokenizor pruning method mentioned in Section 3.2.2, including ablations on tokenizer choice, pruning threshold, etc.
> - **Regarding characterization of our workflowagent + gpt-4o method for WebArena:** to enhance clairty, we  now characterize our approach as a multi-agent system. We restructure Section 4.3 and the ablation studies to emphasize our agent’s contribution to the strong performance.
> - **Regarding result demonstrations:** We have included detailed Mind2Web outputs  in the supplementary material to show the quality of our agent’s outputs. We have also added examples of successful task trajectories for each domain in WebArena to Appendix A 6.4.
>
> &nbsp;
>
> Now, we address your comments individually. We have organized our response by grouping similar concerns together.
>
> **Weaknesses 1: _Novelty. However, this paper deploys conventional and common fine-tuning methods (LoRA) on a widely-used open-source LLM (Qwen2-7B-Instruct) for existing scenarios (LLMs for web tasks), making the paper less innovative._**
>
> We appreciate the feedback but believe this is a reductive characterization of our work. It overlooks the big picture novelty, which is that **WorkflowAgent is the first to show that _a single small-scale, open-source LLM_ can significantly outperform large proprietary models like GPT-4 on web navigation tasks. No previous work has demonstrated such empirical success.** Thus, our work pushes the boundaries of what open-source LLMs can achieve and provides important motivations for future fine-tuning-based research, substantiating its novelty and impact.
>
> While we indeed leverage LoRA, the novelty does not lie in the fine-tuning method per se but in the application and performance improvements we obtain in the context of web agent development. In general, obtaining drastically better results relative to recently published works at top conferences using an intuitive and simple approach is a significant, non-trivial technical contribution [1]. This is also echoed by reviewer 8i3u: *“The authors present an alternative approach that fine-tunes open-source LLMs using production-scale workflow data…An innovative approach for the problem of creating a web agent given a task.”*
>
> Beyond the big picture novelty, we also highlight a few other contributions:
> - We develop and open-source a DOM processing pipeline that balances effectiveness (preserving important DOM information) and efficiency (reducing context window length). This pipeline includes techniques like tag filtering, attribute filtering, and tokenizor pruning and is proven highly effective in practice
> - We perform detailed ablation studies on LLM selection, context window size, and dataset size, which provide valuable empirical insights to future research in the community.
>
> [1] Michael J. Black, Novelty in Science: A Guide for Reviewers, Perceiving Systems Blog
>
> &nbsp;
>
> **Weaknesses 2. _If possible, I kindly ask the authors to consider open-sourcing desensitized datasets or open-sourcing models trained on the newly curated datasets that are used in the experiments for better reproducibility._**
>
> **Question 2. _Would it be possible to open-source the desensitized dataset or only the models trained on the newly curated dataset used in the experiments for better reproducibility?_**
>
> Thank you for recognizing the importance of our large-scale workflow dataset. We understand and appreciate your concerns regarding the accessibility of the proprietary dataset and the model that we can release. To better protect user data, we are actively working on data desensitization. Yet this would be a long-term effort. In the short term, we will contribute to the research community by releasing the complete preprocessing, training, and inference code; evaluation code for external benchmarks; and agents trained on public datasets. We hope these efforts demonstrate our commitment to transparency, and the reviewer could recognize the other contributions our work makes to the field.

---

> ### Author Response · Authors · 2024-11-19
>
> **Weaknesses 3. _Motivation. Considering more and more recent studies in this field have utilized MLLMs rather than text-only LLMs to better capture visual information in web tasks, the motivation for choosing a text-only approach needs to be further clarified._**
>
> **Question 3. _MLLMs can better capture visual information, which can be beneficial in web navigation. What is the motivation for choosing a text-only LLMs-based approach?_**
>
> We agree that incorporating vision models  could help overcome the limitations of context window length and potentially enhance performance. However, we chose not to include a vision component in this work for two reasons:
> - Data Limitations: Our current dataset mostly consists of text-based information. We lack an extensive paired visual dataset necessary to fine-tune a vision model effectively.
> - Focus of the Study: The primary goal of this paper is to demonstrate the advantages of specialized fine-tuning over traditional prompting techniques for web agent development. Introducing an additional modality, such as vision, could complicate the story and detract from the core focus of our research.
>
> Despite these reasons, we agree with the reviewer that developing multi-modal web agents holds promising potential for performance improvements. We intend to explore this in future work. We have added the above clarification to Section 2 of the paper.
>
> &nbsp;
>
> **Weaknesses 4. _Baselines. Some more recent and stronger MLLM web agents also have text-only setting (e.g. WebVoyager_text-only[1]), and they can be compared with the proposed WorkflowAgent as stronger baselines._**
>
> Thank you for the baseline suggestion! We have added a reference to text-only WebVoyager in the related work section. However, we note that the original paper does not provide direct performance results on Mind2Web or WebArena. Therefore, we leave evaluating and comparing WebVoyager with our agent on both external benchmarks and our proprietary dataset as future work.

---

> ### Author Response · Authors · 2024-11-19
>
> **Question 1. _Section 3.2.2 tag attribute filtering: Why are attributes with a character-to-token ratio smaller than 2 considered not semantically meaningful and thus removed? Please elaborate on more details of the intuition here._**
>
> The intuition behind our approach is based on the observation that typical English words consist of more than two characters. Assuming the token count is $t$ and the character count is $s$, this means that when $t=1$, $s\geq 2$, leading to $\frac{s}{t}​ ≥2$. By setting the pruning threshold to 2, we aim to eliminate strings composed solely of single-character tokens, which are likely to be nonsensical.
> Additionally, we add a series of ablation studies to investigate (1) the tokenizer choice, (2) the threshold choice, and (3) patterns of attributes removed. **All results and analysis below can be found in Appendix A 1.2.**
>
> Specifically, we take three tokenizers—Qwen2-7B-Instruct, Mistral-7B-Instruct-v0.3, and Meta-Llama-3-8B—and vary their pruning thresholds to be {1.5, 1.75, 2, 2.25, 2.5}. Note that it is meaningless to study overly small thresholds (e.g.,  it is impossible to have $\frac{s}{t} < 1$) or overly large thresholds (e.g., $\frac{s}{t} < 3$ could result in the loss of meaningful attributes, as many English words contain three letters). We randomly sample 1000 DOMs from our proprietary test dataset, apply our standard pruning pipeline followed by tokenizer pruning, and then perform three analysis:
> - False positives: we use the Python `enchant` library to detect if there are meanful English words within the pruned strings.
> - Average $s$ and $t$ for the entire DOM before and after tokenizer pruning.
> - Lastly, we sort tags and attributes by the frequency of being pruned to identify patterns.
>
> | Tokenizer        | Prune Threshold | False Positive (%) | Before $s$ (K) |Before $t$ (K) | After $s$ (K) |After $t$ (K) |
> |---|---|---|---|---|---|---|
> |Qwen2-7B-Instruct | 1.5| 0.025| 224.3  | 79.14    | 221.4  | 77.11| 2.03|
> |    | 1.75   | 0.013   |   224.3  | 79.14    | 217.3  | 74.67| 4.47|
> |     | 2   | 0.18   | 224.3  | 79.14     | 215.7  | 73.89| 5.21|
> |  | 2.25   | 0.36   | 224.3  | 79.14   | 213.9  | 73.13| 6.01|
> |      | 2.5       | 0.38       |  224.3  | 79.14     | 210.0  | 71.63| 7.51|
> |    Mistral-7B-Instruct-v0.3     | 1.5  | 0.012   | 224.3  | 90.54    | 219.5  | 87.10| 3.44|
> |  | 1.75   | 0.18     |  224.3  | 90.54     | 216.1  | 85.07| 5.47|
> |  | 2   | 0.44  | 224.3  | 90.54   | 212.7  | 83.40| 7.14|
> |    | 2.25   | 0.49   |   224.3  | 90.54     | 205.3  | 80.20| 10.34|
> |     | 2.5   | 11.28    |  224.3  | 90.54     | 190.3  | 74.44| 16.10|
> |      Meta-Llama-3-8B |   1.5             | 0.0097             | 224.3  | 71.44    | 223.1  | 70.60| 0.84|
> |   | 1.75  | 0.012              |   224.3  | 71.44      | 218.3  | 67.85| 3.59|
> |  | 2   | 0.035              |    224.3  | 71.44     | 216.8  | 67.09| 3.43|
> |      | 2.25            | 0.023              |  224.3  | 71.44   | 215.2  | 66.41| 5.03|
> |      | 2.5     | 0.10               |  224.3  | 71.44    | 212.7  | 65.46| 5.98|
>
>
> As shown in the above table, there is a clear trade-off between precision and context reduction: greater reductions in content length tend to result in higher false positive rates. While different tokenizers exhibit varying sensitivities to the pruning thresholds, a threshold of 2 achieves the most balanced trade-off, which aligns with our intuition. **Since the false positive rates for threshold 2 are all quite small, we confirm empirically that our tokenizer pruning method indeed prunes away non-useful attributes consisting of random characters. We have also included a text file demonstrating attributes that are pruned away in our updated supplementary material.**
>
>  We then study the type of attributes by inspecting the top tag-attribute pairs most frequently pruned under threshold 2 along with their  pruning counts:
> - Qwen: (`div`, `class`): 3188, (`span`, `class`): 11426, (`a`, `href`): 8802,   (`button`, `class`): 6844, (`i`, `class`): 5010
> - Mistral: (`div`, `class`): 5288, (`span`, `class`): 15824, (`a`, `href`): 12948,  (`button`, `class`): 7998,  (`svg`, `class`): 5871
> - Llama: (`div`, `class`): 29559, (`span`, `class`): 8823,(`button`, `class`): 5889, (`i`, `class`): 4608,  (`svg`, `class`): 2577
>
> Attributes such as `class` often contain random character strings and are frequently pruned. However, we observe differences in how tokenizers handle the `href` attribute: both Qwen and Mistral tokenizers tend to prune it away, whereas the Llama tokenizer preserves it, indicating its better capability in tokenizing URLs. Although we currently use the Qwen tokenizer in our preprocessing pipeline to align with the backbone model of our agent, the Llama tokenizer can be a compelling alternative for future consideration since it is better at recognizing URLs and producing shorter token sequences.
>
> Please let us know if you have any further questions regarding the pruning method!

---

> ### Author Response · Authors · 2024-11-24
> **Looking forward to feedback on rebuttal**
>
> Thank you again for your valuable feedback! At this moment, we hope you have taken the time to consider our responses to your review. If you have any additional questions or concerns, please let us know so we can resolve them before the discussion period concludes. Otherwise, it would be greatly appreciated if you could raise your score to show that the existing concerns have been addressed. Thank you!

---

> ### Author Response · Authors · 2024-11-26
> **Looking forward to feedback**
>
> Thank you again for your valuable feedback! At this moment, we hope you have taken the time to consider our responses to your review. If you have any additional questions or concerns, please let us know so we can resolve them before the discussion period concludes. Otherwise, it would be greatly appreciated if you could raise your score to show that the existing concerns have been addressed. Thank you!

---

> > ### Comment · Reviewer_Wrtp · 2024-11-27
> >
> > Dear Authors,
> >
> > Thank you for providing a detailed response to my comments and concerns. After carefully reviewing your rebuttal, I have decided to maintain my initial score of 5. This decision is based on the limited technical depth and novelty of the work, as many of the proposed strategies rely on heuristics.

---

### Official Review · Reviewer_8i3u · 2024-11-02

**Soundness:** 3
**Presentation:** 2
**Contribution:** 2
**Rating:** 5
**Confidence:** 3

**Summary:**

The authors present an alternative approach that fine-tunes open-source LLMs using production-scale workflow data collected to develop specialized web agents. They show that they beat GPT4 on the Mind2Web benchmark with a smaller fine-tuned models on their proprietary dataset. They prepare a proprietary dataset which target is to predict the next step given the website’s DOM and action history. They propose an HTML preprocessing strategy that balances between preserving essential information and minimizing context length.

**Strengths:**

- An innovative approach for the problem of creating a web agent given a task
- Using a well-crafted production-scale dataset that contains information about the website's DOM, the action history with the next step as the target the authors show that they can finetune language models with about 7B parameters and outperform much bigger ones. This is done with a parameter efficient method (LoRA)
- They present a DOM preprocessing step with to get rid of useless information to make as much information fit in the LLM given its context length
- If the DOM is not fitting in the model, they chunk it sequentially
- They show that their dataset's training size has an effect on the accuracy

**Weaknesses:**

- There isn't any type of memory component when it comes to DOM. I think that remembering the previous pages will help. But to include something like this it should probably be embedded as a representation due to limitations in the context length
- In table 1 where you show the performance of the fine-tuned LLMs, add another column to show the same model's performance without finetuning to see the performance gain of the fine-tuning. I see that you the performances of non fine-tuned models on your proprietary in table 4 which makes me wonder why you didn't also include in table 1. dataset baseline
- You assume that the non-useful attributes are the ones which have character to token ratio > 2. This may not always be the case and is highly dependent on the tokenizer and the type of attribute.

**Questions:**

- The performance in table1 is for the Mind2Web or WebArena? I don't see it mentioned somewhere
- "For performance robustness, we call WorkflowAgent five times and use majority vote to select the final answer". Do you use sampling when you generate the action? What is the reason behind it? Is following a self-consistency approach?
- Is there any reason to not approach this problem with a vision model? You would get rid of many difficulties like preprocessing or making sure the page fits in the model. It would be interesting to train this with a Vision model and actual screenshots of the pages instead of the HTML of the page and evaluate on public benchmarks like multi-modal mind2web. This would make it more flexible and you would not need to preprocess to remove redundant information. Also incorporate previous pages as a context as a form of “memory”

---

> ### Author Response · Authors · 2024-11-19
>
> Thank you for your constructive feedback. Before addressing each of your comments individually, we would like to outline key revisions made in the rebuttal:
> - **Regarding data source:** We have added the following clarification to the paper: “our dataset is collected using a public software tool designed to streamline the creation of step-by-step guides. It allows users to record their interactions with the web through a browser extension and converts the interactions into well-annotated, step-by-step instructions, which can then be customized to   specific business needs.” Since the software tool is closely tied to our institution, we cannot disclose its name in order to maintain anonymity and adhere to double-blind review policy. However, to provide a clearer understanding of our dataset, we have included raw data from an example workflow performed by the authors in the supplementary material.
> - **Regarding the pruning method in DOM preprocessing:** In Appendix A 1.2, we have added detailed analysis about the tokenizor pruning method mentioned in Section 3.2.2, including ablations on tokenizer choice, pruning threshold, etc.
> - **Regarding characterization of our workflowagent + gpt-4o method for WebArena:** to enhance clairty, we  now characterize our approach as a multi-agent system. We restructure Section 4.3 and the ablation studies to emphasize our agent’s contribution to the strong performance.
> - **Regarding result demonstrations:** We have included detailed Mind2Web outputs  in the supplementary material to show the quality of our agent’s outputs. We have also added examples of successful task trajectories for each domain in WebArena to Appendix A 6.4.
>
> &nbsp;
>
> Now, we address your comments individually. We have organized our response by grouping similar concerns together.
>
> **Weaknesses 1: _There isn't any type of memory component when it comes to DOM. I think that remembering the previous pages will help. But to include something like this it should probably be embedded as a representation due to limitations in the context length._**
>
> **Question 3: _Also incorporate previous pages as a context as a form of “memory”_**
>
> Thank you for the suggestion. We totally agree that adding memory components can enhance the agent’s ability to manage long contexts and multi-step interactions.
> While we recognize this limitation in the current system, the primary focus of this work has been to demonstrate the advantages of fine-tuning with specialized data over prompting proprietary models. For both our agent and the baseline proprietary models, we do not include memory components to ease implementation and ensure fair comparison.
> Thus,  we leave integrating  memory components as a future work.
>
> Note that we explicitly mentioned in the conclusion (Section 5) that ``we aim to enable Workflow to compare and reason over multiple DOM chunks so that its observation is always complete. This might require integrating a memory component, which could also aid in maintaining context or state across interactions to improve multi-step reasoning.” We have also discussed relevant literature that focuses on memory integration in the related work section, highlighting studies such as AWM and Synapse.

---

> ### Author Response · Authors · 2024-11-19
>
> **Weaknesses 2: _In table 1 where you show the performance of the fine-tuned LLMs, add another column to show the same model's performance without finetuning to see the performance gain of the fine-tuning._**
>
> Thank you for the suggestion. We have added results before fine-tuning to Table 1 in the revised manuscript. Now the table looks like:
>
> | Model                     | # Params   | Before Fine-Tuning |                 | After Fine-Tuning |                 |
> |---------------------------|------------|--------------------|-----------------|-------------------|-----------------|
> |                           |            | EM (%)             | Calibrated EM (%) | EM (%)          | Calibrated EM (%) |
> | Mistral-7B-Instruct-v0.3  | 7B         | 3.89               | 5.13            | 19.92             | 26.31            |
> | Qwen2-7B-Instruct         | 7B         | 6.06               | 7.92            | 29.34             | 38.72            |
> | Llama-3.1-Instruct-8B     | 8B         | 1.42               | 1.88            | 28.34             | 37.42            |
> | Qwen2.5-14B-Instruct      | 14B        | 8.79               | 11.6            | 31.76             | 41.89            |
> | Codestral-22B-v0.1        | 22B        | 4.53               | 6.08            | 31.11             | 41.25            |
> | Mixtral-8x7B-Instruct-v0.1| 56B-A12B   | 7.35               | 9.82            | 28.38             | 37.49            |
> | Qwen2-57B-A14-Instruct    | 57B-A14B   | 5.72               | 7.51            | 31.02             | 40.10            |
>
> The takeaway is that without fine-tuning, open-source LLMs tend to perform poorly on web-based tasks since they are not trained to process HTML. However, after our specialized fine-tuning, there is a significant improvement in performance, clearly demonstrating the effectiveness of our approach.

---

> ### Author Response · Authors · 2024-11-19
>
> **Weaknesses 3: _You assume that the non-useful attributes are the ones which have character to token ratio > 2. This may not always be the case and is highly dependent on the tokenizer and the type of attribute._**
>
> Thank you for your suggestion. To motivate our tokenizer pruning scheme, we add a series of ablation studies to investigate (1) the tokenizer choice, (2) the threshold choice, and (3) patterns of attributes removed. **All results and analysis below can be found in Appendix A 1.2.**
>
> Specifically, we take three tokenizers—Qwen2-7B-Instruct, Mistral-7B-Instruct-v0.3, and Meta-Llama-3-8B—and vary their pruning thresholds to be $\{1.5, 1.75, 2, 2.25, 2.5\}$. Note that it is meaningless to study overly small thresholds (e.g.,  it is impossible to have $\frac{s}{t} < 1$) or overly large thresholds (e.g., $\frac{s}{t} < 3$ could result in the loss of meaningful attributes, as many English words contain three letters). We randomly sample 1000 DOMs from our proprietary test dataset, apply our standard pruning pipeline followed by tokenizer pruning, and then perform three analysis:
> - False positives: we use the Python `enchant` library to detect if there are meanful English words within the pruned strings.
> - Average $s$ and $t$ for the entire DOM before and after tokenizer pruning.
> - Lastly, we sort tags and attributes by the frequency of being pruned to identify patterns.
>
> | Tokenizer        | Prune Threshold | False Positive (%) | Before $s$ (K) |Before $t$ (K) | After $s$ (K) |After $t$ (K) |
> |---|---|---|---|---|---|---|
> |Qwen2-7B-Instruct | 1.5| 0.025| 224.3  | 79.14    | 221.4  | 77.11| 2.03|
> |    | 1.75            | 0.013   |   224.3  | 79.14    | 217.3  | 74.67| 4.47|
> |     | 2               | 0.18   | 224.3  | 79.14     | 215.7  | 73.89| 5.21|
> |  | 2.25            | 0.36   | 224.3  | 79.14   | 213.9  | 73.13| 6.01|
> |      | 2.5             | 0.38       |  224.3  | 79.14     | 210.0  | 71.63| 7.51|
> |    Mistral-7B-Instruct-v0.3     | 1.5  | 0.012   | 224.3  | 90.54    | 219.5  | 87.10| 3.44|
> |                  | 1.75            | 0.18               |  224.3  | 90.54     | 216.1  | 85.07| 5.47|
> |                  | 2               | 0.44               | 224.3  | 90.54   | 212.7  | 83.40| 7.14|
> |                  | 2.25            | 0.49           |   224.3  | 90.54     | 205.3  | 80.20| 10.34|
> |     | 2.5             | 11.28              |  224.3  | 90.54     | 190.3  | 74.44| 16.10|
> |      Meta-Llama-3-8B              | 1.5             | 0.0097             | 224.3  | 71.44    | 223.1  | 70.60| 0.84|
> |                  | 1.75            | 0.012              |   224.3  | 71.44      | 218.3  | 67.85| 3.59|
> |                  | 2               | 0.035              |    224.3  | 71.44     | 216.8  | 67.09| 3.43|
> |                  | 2.25            | 0.023              |  224.3  | 71.44   | 215.2  | 66.41| 5.03|
> |                  | 2.5             | 0.10               |  224.3  | 71.44    | 212.7  | 65.46| 5.98|
>
>
> As shown in the above table, there is a clear trade-off between precision and context reduction: greater reductions in content length tend to result in higher false positive rates. While different tokenizers exhibit varying sensitivities to the pruning thresholds, a threshold of 2 achieves the most balanced trade-off, which aligns with our intuition. **Since the false positive rates for threshold 2 are all quite small, we confirm empirically that our tokenizer pruning method indeed prunes away non-useful attributes consisting of random characters. We have also included a text file demonstrating attributes that are pruned away in our updated supplementary material.**
>
>  We then study the type of attributes by inspecting the top tag-attribute pairs most frequently pruned under threshold 2 along with their  pruning counts:
> - Qwen: (`div`, `class`): 3188, (`span`, `class`): 11426, (`a`, `href`): 8802,   (`button`, `class`): 6844, (`i`, `class`): 5010
> - Mistral: (`div`, `class`): 5288, (`span`, `class`): 15824, (`a`, `href`): 12948,  (`button`, `class`): 7998,  (`svg`, `class`): 5871
> - Llama: (`div`, `class`): 29559, (`span`, `class`): 8823,(`button`, `class`): 5889, (`i`, `class`): 4608,  (`svg`, `class`): 2577
>
> Attributes such as `class` often contain random character strings and are frequently pruned. However, we observe differences in how tokenizers handle the `href` attribute: both Qwen and Mistral tokenizers tend to prune it away, whereas the Llama tokenizer preserves it, indicating its better capability in tokenizing URLs. Although we currently use the Qwen tokenizer in our preprocessing pipeline to align with the backbone model of our agent, the Llama tokenizer can be a compelling alternative for future consideration since it is better at recognizing URLs and producing shorter token sequences.
>
> Please let us know if you have any further questions regarding the pruning method!

---

> ### Author Response · Authors · 2024-11-19
>
> **Question 1: _The performance in table1 is for the Mind2Web or WebArena?_**
>
> Neither. Table 1 is evaluating on the test split of the dataset we collected. We have added this clarification to the caption.
>
> &nbsp;
>
> **Question 2: _“For performance robustness, we call WorkflowAgent five times and use majority vote to select the final answer". Do you use sampling when you generate the action? What is the reason behind it? Is following a self-consistency approach?_**
>
> Yes, we already use sampling at each time when generating an action with temperature 0.6. Performing multiple inference and using majority vote  is a double check to further enhance output robustness and reduce noise. It is indeed a form of self-consistency check and allows us to better analyze the outputs. While this does increase some computational costs, it improves the model’s ability to generate consistent actions, so we leverage it for Mind2Web.
>
> &nbsp;
>
> **Question 3: _Is there any reason to not approach this problem with a vision model?_**
>
> We agree that incorporating vision models  could help overcome the limitations of context window length and potentially enhance performance. However, we chose not to include a vision component in this work for two reasons:
> - Data Limitations: Our current dataset mostly consists of text-based information. We lack an extensive paired visual dataset necessary to fine-tune a vision model effectively.
> - Focus of the Study: The primary goal of this paper is to demonstrate the advantages of specialized fine-tuning over traditional prompting techniques for web agent development. Introducing an additional modality, such as vision, could complicate the story and detract from the core focus of our research.
>
> Despite these reasons, we agree with the reviewer that developing multi-modal web agents can be fruitful and impactful. We intend to explore this in future work.

---

> ### Author Response · Authors · 2024-11-24
> **Looking forward to feedback on rebuttal**
>
> Thank you again for your valuable feedback! At this moment, we hope you have taken the time to consider our responses to your review. If you have any additional questions or concerns, please let us know so we can resolve them before the discussion period concludes. Otherwise, it would be greatly appreciated if you could raise your score to show that the existing concerns have been addressed. Thank you!

---

> ### Author Response · Authors · 2024-11-26
> **Looking forward to feedback**
>
> Thank you again for your valuable feedback! At this moment, we hope you have taken the time to consider our responses to your review. If you have any additional questions or concerns, please let us know so we can resolve them before the discussion period concludes. Otherwise, it would be greatly appreciated if you could raise your score to show that the existing concerns have been addressed. Thank you!

---

> > ### Comment · Reviewer_8i3u · 2024-11-28
> >
> > Thanks for your responses. I will keep my current score as I think there is not enough value for the research community from your work

---

### Note · Authors · 2024-12-08

I have read and agree with the venue's withdrawal policy on behalf of myself and my co-authors.